



# Microstructure, Micro-inclusions and Mineralogy along the EGRIP ice core - Part 2: Implications for paleo-mineralogy

Nicolas Stoll[1], Maria Hörhold[1], Tobias Erhardt[1, 2], Jan Eichler[1], Camilla Jensen[2], and Ilka Weikusat[1, 3]

[1]Alfred Wegener Institute Helmholtz Centre for Polar and Marine Research, Bremerhaven, Germany
[2]Climate and Environmental Physics, Physics Institute and Oeschger Centre for Climate Change Research, University of Bern, Bern, Switzerland
[3]Department of Geosciences, Eberhard Karls University, Tübingen, Germany

**Correspondence:** Nicolas Stoll (nicolas.stoll@awi.de)

**Abstract.** Impurities in polar ice do not only allow the reconstruction of past atmospheric aerosol concentration, but also influence the physical properties of the ice. However, the mineralogy and location of impurities in ice and the involved processes are poorly understood. We use Continuous Flow Analysis to derive the dust particle concentration and optical microscopy and Cryo-Raman spectroscopy to systematically locate and analyse the mineralogy of micro-inclusions in situ inside eleven solid

ice samples from the upper 1340 m of the East Greenland Ice Core Project ice core. Micro-inclusions are more variable in mineralogy than previously observed and are mainly composed of mineral dust (quartz, mica and feldspar) and sulphates (mainly gypsum). Inclusions of the same composition tend to cluster, but clustering frequency and mineralogy changes considerably with depth. A variety of sulphates dominate the upper 900 m while gypsum is the only sulphate in deeper samples, which however contain more mineral dust, nitrates and dolomite. The analysed part of the core can thus be divided into two depth

regimes of different mineralogy, and to a lesser degree of spatial distribution, which could originate from different chemical reactions in the ice or large-scale changes of ice cover in NE-Greenland during the Mid-Holocene. The complexity of impurity mineralogy on the metre- and centimetre-scale in polar ice is still underestimated and new methodological approaches are necessary to establish a comprehensive understanding of the role of impurities.

## 1 Introduction

Deep ice cores from the polar ice sheets are a valuable archive of information: studying ice crystals allows the analysis of ice dynamics (e.g., Thorsteinsson et al., 1997; Faria et al., 2014; Fitzpatrick et al., 2014) while impurities and isotopes preserved in the ice enable the reconstruction of the paleoclimate of our planet (e.g., Dahl-Jensen et al., 2013; EPICA Community Members, 2004). Soluble impurities are chemical compounds, such as $Na^+$, $Cl^-$, and $NO_3^-$, and chemical compounds of

atmospheric, marine, terrestrial or biological origin which dissolve in the lattice (Legrand and Delmas, 1988). They can also originate from salts dissociated into ions (Legrand and Mayewski, 1997; Della Lunga et al., 2014) or from dissolved gases such





as hydrogen peroxide ($H_2O_2$). Insoluble impurities are rejected from the ice lattice, because they consist of lattice-incoherent phases (Ashby, 1969; Alley et al., 1986). They range in size from sub-micrometre to hundreds of micrometres (Steffensen, 1997; Wegner et al., 2015; Simonsen et al., 2019). Micro-metre sized inclusions are called "micro-inclusions" and the focus of

this study. Insoluble impurities normally originate from terrestrial sources and mainly consist of mineral dust which is abundant in elements from the crust, such as Si, Al, Ca, and Fe, and in chemical compounds, such as $FeSiO_3$, $CaSiO_3$, FeS, and $SiO_2$.

Impurities have different transport histories and originate from different atmospheric aerosols, in the form of e.g., salt particles or terrestrial dust (Legrand and Mayewski, 1997; Weiss et al., 2002), and are deposited on ice sheets. They form solid micro-inclusions or dissolve in the ice lattice or in grain and subgrain boundaries as snow eventually transforms into ice. With

time ice, and the impurities within, move into deeper parts of the ice column. Though the absolute concentrations are extremely low, the impurities inside the polar ice sheets are of interest for various research fields and applications. Tephra and sulphate layer are used for absolute dating and the isotopic and chemical composition of impurities enable the investigation of climatic and atmospheric processes of the past.

Impurities in polar ice exist as soluble (in water or ice) or insoluble inclusions (e.g., dust particles, salts or droplets) and can

be measured with a variety of methods, each with certain limitations. Main differences are the state of the analysed particles (soluble or insoluble in water or ice) and the applied aggregate phase of the sample: methods analysing a liquid and thus melting the ice are e.g., Ion Chromatography (IC)(Cole-Dai et al., 2006), CFA (Röthlisberger et al., 2000a), and Inductively Coupled Plasma Mass Spectrometry (McConnell et al., 2002; Erhardt et al., 2019) while Laser Ablation Inductively Coupled Plasma Mass Spectrometry (LA-ICP-MS) (Reinhardt et al., 2001) and Cryo-Raman spectroscopy (Fukazawa et al., 1998) analyse solid

ice samples.

The impurity content plays a major role regarding the physical properties of snow, firn, and ice, such as electrical conductivity (Alley and Woods, 1996; Wolff et al., 1997), permittivity (Wilhelms et al., 1998), and mechanical properties (e.g., Jones and Glen, 1969; Dahl-Jensen and Gundestrup, 1987; Paterson, 1991; Hörhold et al., 2012). Another important area is the impact of impurities on the deformation of ice (e.g., Jones and Glen, 1969; Petit et al., 1987; Iliescu and Baker, 2008; Eichler et al.,

2019), which in turn influences the flow of ice - a major uncertainty regarding future projections of ice sheet behaviour and solid ice discharge. This chain of processes at the micro-scale resulting eventually in the large-scale deformation behaviour of ice needs to be better understood, especially in fast flowing ice, as present in ice streams. The microstructure of ice can be impacted directly by processes related to impurities such as Zener pinning (Smith, 1948; Humphreys and Hatherly, 2004) or grain boundary drag (Alley et al., 1986, 1989). These result in changes of the energy, shape and mobility of grain boundaries.

Furthermore, impurities such as micro-inclusions directly impact deformation when they form obstacles in the ice matrix and thus enhance strain localisation and the introduction of protonic defects (Glen, 1968) or the formation of dislocation lines (Weertman and Weertman, 1992). In turn, these small-scale processes can also influence climate proxies by altering the stratigraphic integrity of impurity records (Faria et al., 2010; Ng, 2021). Therefore, understanding the localisation of impurities is important.

Over the last decades different ice cores from Greenland and Antarctica were analysed regarding the chemistry and location of impurities and the main results were recently summarised by Stoll et al. (2021). While studies using optical microscopy



by e.g., Kipfstuhl et al. (2006); Faria et al. (2010) focused mainly on the location of insoluble particles Scanning Electron Microscope (SEM) coupled with energy dispersive X-ray spectroscopy (EDS) (e.g., Wolff et al., 1988; Barnes et al., 2002; Baker et al., 2003; Barnes et al., 2003a) and LA-ICP-MS (e.g., Reinhardt et al., 2001; Della Lunga et al., 2014; Bohleber et al., 2020) studies also allowed the identification of impurities. They mainly found chlorine, sulphur and sodium at several locations, ranging from grain boundaries to filaments and the grain interior. However, these methods are limited to the investigation of elemental concentrations. Identifying the mineralogy of inclusions is however possible with Raman spectroscopy (e.g., Fukazawa et al., 1997; Ohno et al., 2005; Sakurai et al., 2011; Ohno et al., 2014; Eichler et al., 2019) while preserving the microstructural location , a crucial aspect regarding their impact on the physical properties of the host material. Identified inclusions in Antarctic and Greenlandic ice mainly consist of terrestrial dust (e.g., quartz, feldspar, mica, hematite) and sulphates of different chemical composition (Ca, Na, Mg, K, Al) (Ohno et al., 2005; Sakurai et al., 2009, 2011; Ohno et al., 2014; Eichler et al., 2019). Calcium carbonate ($CaCO_3$) was identified by Sakurai et al. (2009) mainly in the last glacial period from the Greenland Ice Core Project (GRIP) ice core. All these studies are demanding in time and costly and thus often rely on a small number of samples taken at arbitrary depths. This results in a poor understanding of the underlying processes and in a lack of identified species involved hampering the development of realistic generalisations of impurities for polar ice (Stoll et al., 2021).

In summary, commonly used methods, developed to derive (paleo-) climate information on impurities such as Continuous Flow Analysis (CFA) (e.g., Kaufmann et al., 2008), apply their analysis on melted sample water for the sake of decontamination and time constraints. Doing so, information on the phase and localisation of impurities are not considered. On the other hand, dedicated studies on localisation are spatially very limited and do not catch the overall chemistry of a certain sample. However, in order to develop a profound understanding on the mineralogy and localisation of impurities in polar ice, a combination of both approaches is needed.

Our approach is a combination of methods from microstructure and impurity research to perform a systematic high-resolution analysis along one deep ice core. The East Greenland Ice Core Project (EGRIP) ice core, the first ice core drilled through the Northeast Greenland Ice Stream (NEGIS), enables such an interdisciplinary undertaking. A companion study investigates the location of micro-inclusions and their role regarding deformation and microstructure. In this study we investigate the mineralogy of these visible micro-inclusions. We apply bulk chemistry from CFA and optical microscopy and Raman spectroscopy on eleven samples along the EGRIP ice core. With this combined data set covering the Holocene and Late Glacial, i.e. the upper 1340 m, we aim to give a systematic overview of the evolution of the mineralogy of the EGRIP ice core by identifying a relevant number of micro-inclusions at equal depth intervals. We discuss the interpretation of the mineralogy of micro-inclusions in the EGRIP ice core with respect to environmental boundary conditions and survey implications for future research.

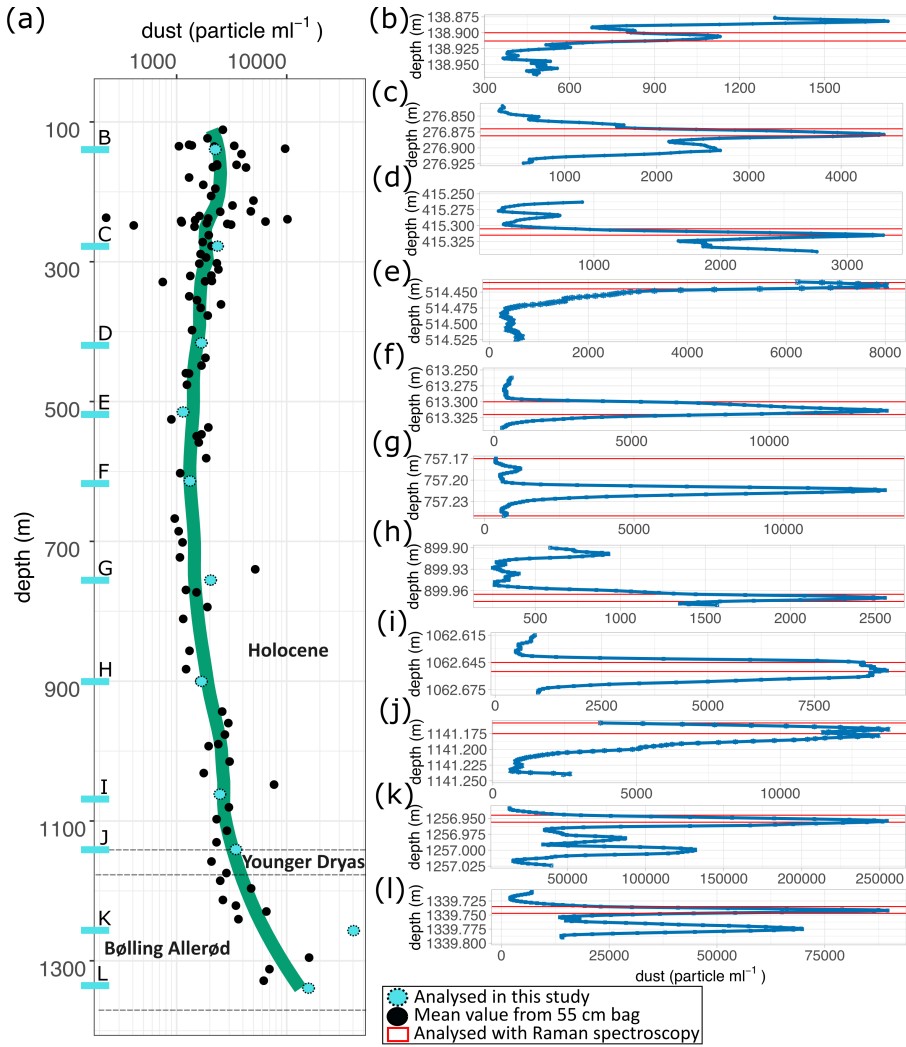

**Figure 1.** Dust data derived via CFA from the upper 1340 m of the EGRIP ice core and depth regimes analysed with Raman spectroscopy. A) Mean dust particle concentrations of 55 cm bags. The green line is a locally weighted regression with a smoothing parameter of 0.3. B)-L) CFA dust data from the chosen depths analysed with Raman spectroscopy. The concentration range on the abscissa varies. Sample K is within a cloudy band.

## 2 Methods

### 2.1 The East Greenland Ice Core Project

The EGRIP ice core drilling project is located at 75°38' N and 35°58' W, 2704 m a.s.l. (2015) on NEGIS, the largest ice

stream in Greenland (Joughin et al., 2010; Vallelonga et al., 2014). At the drill site the ice flows with a velocity of 55 m $a^{-1}$ (Hvidberg et al., 2020) offering a unique possibility to study ice rheology and physical parameters contributing to deformation





such as crystal preferred orientation and impurity content. While first results regarding the physical properties of EGRIP ice were recently published (Westhoff et al., 2020) we investigate microstructure and micro-inclusions in detail in a companion paper.

The present day accumulation in water equivalent at the EGRIP drill site is between 13.8 and 14.9 cm/yr (Nakazawa et al., 2021) and the averaged annual layer thicknesses for the period 1607–2011 is 0.11 m ice equivalent (Vallelonga et al., 2014). In the last 8 kyr before 2000 CE annual layer thicknesses were almost constant, probably due to a combination of flow-induced thinning and increased upstream accumulation (Mojtabavi et al., 2020). Traced radar layers from the North Greenland Ice Core Project (NGRIP) drill site indicate an undisturbed climatic record of at least 51 kyr (Vallelonga et al., 2014).

Drilling at EGRIP has not continued since 2019 and the current drill depth is 2121 m, roughly 530 m above bedrock. We analyse the upper 1340 m of the core covering the Holocene (present-11.7 ka) in the upper 1240 m, the Younger Dryas (11.7-12.8 ka) at 1240-1280 m, and the Bølling Allerød (12.8-14.7 ka) at 1280-1375 m (Walker et al., 2018; Mojtabavi et al., 2020).

## 2.2    Continuous Flow Analysis

CFA data were obtained during three measurement campaigns (2018-2020) at the University of Bern using the CFA setup described in detail in Kaufmann et al. (2008) with a few minor improvements. For the analysis a longitudinal 35 x 35 mm section of the ice core was cut and melted down-core and analysed using continuous measurement methods to produce mm-cm resolution records of soluble and insoluble impurities in the ice. Micro-particle concentrations where determined using an Abakus (Fa Klotz) Laser Particle Sizer (e.g., Ruth et al., 2003) operating in the size range between 1-15 $\mu$m, which covers the

size range of optical microscopy. The depth resolution of the data is limited by signal dispersion in the analytical system to approximately 0.5-1 cm. Depth co-registration to the thin sections is limited by the accuracy of the depth assignment in the field and is typically on the order of a few mm. We calculated the arithmetic mean dust particle concentration of each bag for an overview.

## 2.3    Sample preparation

Aiming to derive a systematic overview over the EGRIP core we selected samples roughly every 100 m of depth. Samples were chosen based on a combination of depth-representative and prominent features in the CFA (e.g., high insoluble particle concentration), grain size (e.g., fine) and crystal preferred orientation data. The latter two are presented in a companion paper, we here focus on the chemistry. Exact regions of interests in these samples for microstructural and Raman analysis were defined using high-resolution CFA data (Fig. 1). The chosen areas included areas without prominent properties and specific areas of

interest, such as high dust content. We analysed eleven samples between depths of 138.92 and 1339.75 m, taken every ~110 m (Fig. 1 and Table 1). The nine shallower samples are from the Holocene and the two deepest from the last glacial termination (Mojtabavi et al., 2020).

As described in a companion paper we used the remaining ice of the physical properties samples analysed at the EGRIP camp and created thick sections following a standard procedure (Kipfstuhl et al., 2006). No silicon oil was used to avoid ar-



**Table 1.** Properties of the analysed samples.

| Depth (m) | Age b2k (ka) | Size (mm x mm) |
|---|---|---|
| 138.92 | 1.0 | 13.01 x 16.96 |
| 276.88 | 2.2 | 11.00 x 12.76 |
| 415.30 | 3.5 | 10.01 x 13.27 |
| 514.44 | 4.3 | 9.94 x 60.60 |
| 613.30 | 5.2 | 17.31 x 54.19 |
| 757.21 | 6.4 | 90.00 x 16.78 |
| 899.98 | 7.6 | 11.32 x 12.62 |
| 1062.65 | 9.3 | 9.67 x 8.47 |
| 1141.17 | 10.2 | 16.61 x 21.67 |
| 1256.96 | 12.1 | 10.45 x 48.50 |
| 1339.75 | 14.1 | 11.24 x 11.96 |

b2k=before 2000 CE (Mojtabavi et al., 2020)

tificial Raman spectra masking the micro-inclusion spectra. Most samples were approximately 10 x 10 mm, but dimensions varied depending on the CFA data (Table 1). A Leica microtome was used to polish the top and bottom surface of the samples, which was followed by 1.5-2 hours of sublimation under controlled humidity and temperature conditions. Thus, a good sample surface quality was obtained and small-scale disturbances (e.g., microtome scratches) were erased while grain-boundary grooves became more distinct. Polished surfaces allow the localisation of micro-inclusions 500 $\mu$m below the sample surface

and successful Raman spectroscopy measurements with strong signals.

### 2.4 Microstructure mapping and impurity maps

Microstructure mapping was performed following Kipfstuhl et al. (2006) and Eichler et al. (2017) and the detailed investigation of the microstructural location of micro-inclusions is explained in detail in a companion paper. Samples were placed under an optical Leica DMLM microscope with an attached CCD camera (Hamamatsu C5405) and several hundred individual pho-

tomicrographs were created grid-wise with a scanning resolution of 3 $\mu$mpix$^{-1}$. The creation of high-resolution maps to detect micro-inclusions provided the basis for a structured Raman analysis. These inclusions are usually 1-2 $\mu$m in diameter (Fig. 3A) and consist of droplets, dust particles or salts. We focused inside the ice using transmission light mode and different focus depths to locate micro-inclusions below the surface (Fig. 2B). Microstructure maps with indicated micro-inclusions are from now on called "impurity maps". We mapped grain boundaries on the sample surface and translated them into the impurity map

(Fig. 2) by applying a grain boundary width of 300 $\mu$m. This is an upper-limit assumption and compensates for vertically tilted grain boundaries and internal light diffraction with depth (Eichler et al., 2017). Micro-inclusions inside these grain boundaries were classified as "at the grain boundary" (Fig. 2)C.

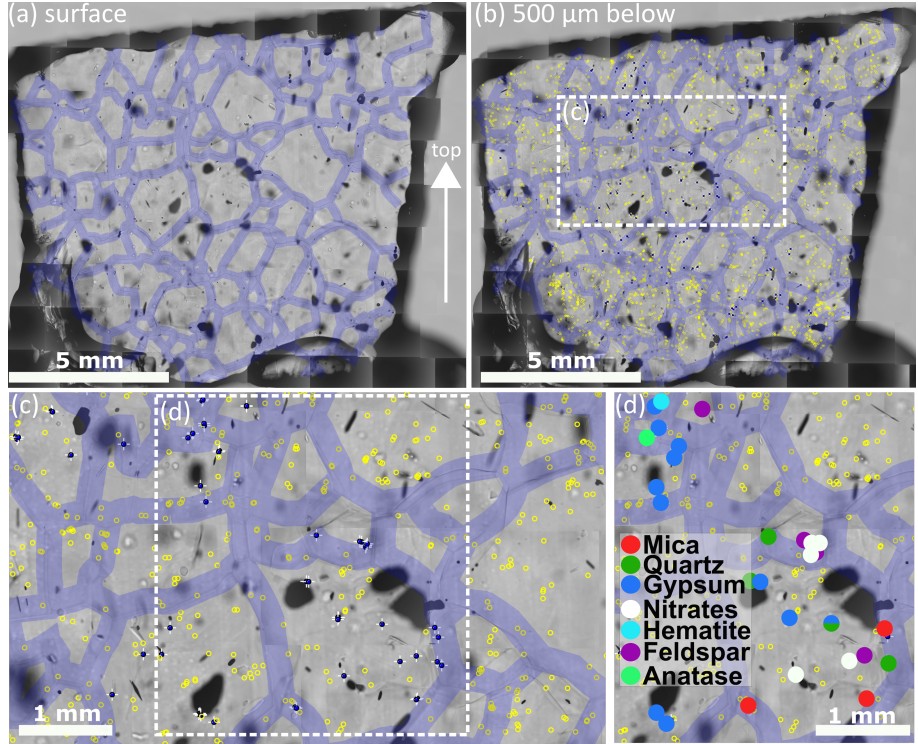

**Figure 2.** Details of the analysis procedure on the sample from a depth of 1339.75 m. Grain boundaries are indicated with 300 $\mu$m thick violet lines. Localised micro-inclusions 500 $\mu$m below the surface are indicated by yellow circles; micro-inclusions analysed with Raman spectroscopy are indicated by filled blue circles with white crosses. Out of focus black shapes are air bubbles. A) Map of the sample surface with highlighted grain boundaries, the arrow indicates the surface of the ice sheet. B) Impurity map with a focus depth of 500 $\mu$m below the sample surface, micro-inclusions and grain boundaries are indicated. C) Detail of the area indicated in B. D) Identified Raman spectra of the micro-inclusions indicated in C. One inclusion consists of a quartz and gypsum spectra simultaneously.

## 2.5 Cryo-Raman spectroscopy

The Raman effect is the inelastic scattering of light caused by the excitation of vibrational modes of crystals or molecules.
This results in the Raman shift – a loss of scattered light energy, which is specific for each vibrational mode. The Raman shift is used to identify chemical impurities in samples in a non-destructive way – Raman spectroscopy. It is well suited for light-transparent materials, such as ice. Using confocal optics, it is possible to focus the excitation laser on a small volume segment (few $\mu m^3$) inside the sample (Fig. 3A). The same confocal optics collect the backscattered light, which is conducted into a spectrometer to resolve the spectral distribution. Due to the low probability of a photon being Raman-scattered long
integration-times, high incident light intensities and sufficient particle concentration is needed. To avoid melting, laser power is limited and thus, well-prepared sample surfaces and precise focusing of the laser is essential to obtain identifiable spectra (Fig. 3B-D).

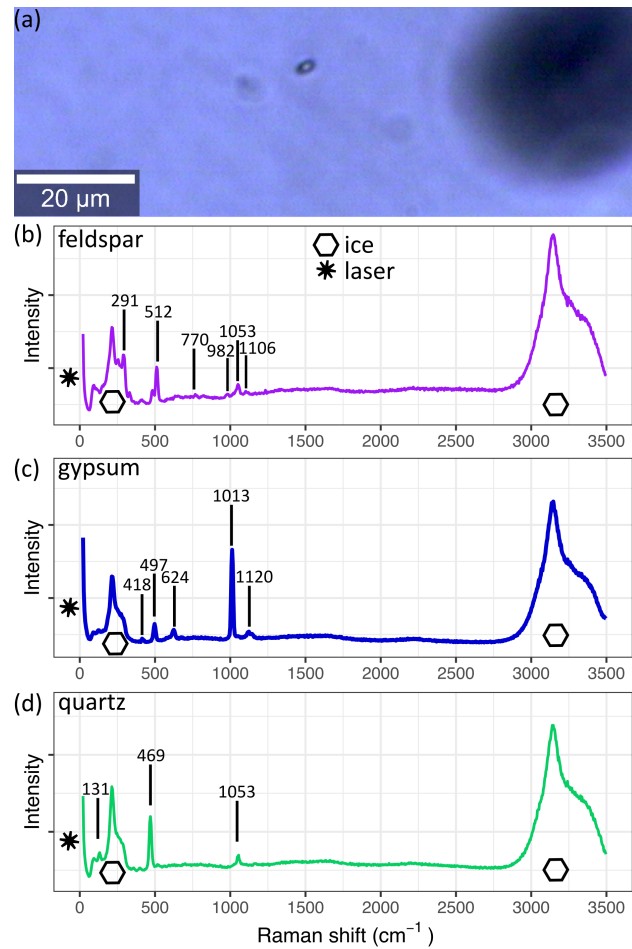

**Figure 3.** A micro-inclusion at 1339.75 m of depth as seen with the 20x lens of the Raman system. The derived spectrum of the inclusion is shown in B). The spectra in C) and D) are from the same sample and frequently measured throughout the core. Superimposed ice and laser signals are indicated. A) The small black object is a micro-inclusion, the larger black object out of focus is an air bubble. B) Observed spectra of feldspar. C) Observed spectra of gypsum. D) Observed spectra of quartz.

When using Raman spectroscopy on micro-inclusions inside ice, the ice spectrum is always present (Fig. 3B-D). Other factors hampering clear Raman signals are: 1) small sizes of micro-inclusions, 2) laser beam path length, 3) quality of sample surface, and 4) acquisition time. Normally, one micro-inclusion was measured ten times with an acquisition time of 0.5-2 s per spectrum, resulting in a total acquisition time of at least 5 s. Weak signals were remeasured with up to 3 s per spectrum and thus, up to 30 s in total. The potential cumulative heating of the micro-inclusion limits the total acquisition time, because it could result in melting of the sample or the destruction of the inclusion.

We measured in a maximal focus depth of 500 $\mu$m below the surface to obtain high-quality signals (Fig. 2B). Surface pollution and possible reactions or rearrangements of elements on the surface were avoided by focusing into the sample, thus





only analysing inclusions in the ice. Nevertheless, this leads to a permanent superimposition of the ice spectrum. Impurity peaks were well distinguishable in most cases, even though some signals with strong ice-vibrational bands were hidden by the Raman-active ice spectra. This is often the case for O-H signals of hydrates, which are luckily shifted to higher frequencies in minerals and normally very strong – distinguishing them was usually possible.

Spectroscopy analysis was performed at the Alfred Wegener Institute Helmholtz Centre for Polar- and Marine Research, Bremerhaven with a WITec alpha 300 M+ combined with a NdYAG laser ($\lambda = 532\,\mathrm{nm}$) and a UHts 300 spectrometer with a 600 grooves mm⁻¹ grating. The used Raman system is further described by Weikusat et al. (2015). The microscope unit is located in a small cold lab inside an insulated cabin with a temperature of -15°C. The control unit, spectrometer and excitation laser are located close to the cell at room temperature and connected with the microscope. We used a 100 $\mu m$ fibre to obtain a good

compromise between signal intensity and confocality. If necessary, samples were re-polished in the same cabin, allowing fast and controlled sample sublimation and a low risk of contamination. The small cabin volume resulted in increasing temperatures over time; measurements were stopped at the very latest at -8°C. To avoid extensive sublimation measurements were continued on the next day. Sample analysis was carried out until a maximum of ~500 $\mu m$ from the sample surface sublimated. However, we were able to analyse high numbers of micro-inclusions, often exceeding 100 micro-inclusion measurements per sample.

Raman spectra were background corrected and identified using reference spectra (e.g., Ohno et al., 2005; Eichler et al., 2019), and the RRUFF project database (Lafuente et al., 2015) (examples in 3B-D).

## 3    Results

### 3.1    Evolution of insoluble particle number with depth

To put the samples analysed for micro-inclusions into a broader context, Fig. 1A shows a profile of 55 cm average micro-

particle concentrations over the entire depth range of the core. Generally, during the Holocene, particle concentrations are relatively low, compared to the Preboral and Younger Dryas periods in good agreement with previous studies of micro-particle concentrations ins Greenland ice (Steffensen, 1997; Ruth et al., 2003). However, a detailed climatological interpretation of the data, including e.g., possible upstream and effects, is outside the scope of this study and will follow at a later time.

     Mean particle numbers are generally consistent and vary most at a depth of 250 m. The general increase in insoluble particle

numbers with depth also occurs in our chosen samples and is up to two orders of magnitude from the shallowest to the deepest samples (Fig. 1B-L).

     The four shallowest samples show insoluble particle peaks between 1400 and 8000 $ml^{-1}$ (Fig. 1B-E), while the highest values are observed in the three deepest samples (up to >250000 $ml^{-1}$) (Fig. 1J-L). There is a high variability of the insoluble particle concentration on small spatial scales and significant changes occur within centimetres due to the pronounced season-

ality in the dust deposition in Greenland. Fig. 1K) and 1L) display insoluble particle numbers skyrocketing and dropping over a few centimetres.



**Table 2.** Most abundant Raman spectra in EGRIP Holocene ice and localisation data. Sulphates, except gypsum, at grain boundaries were not distinguished and are indicated with hyphens.

| Mineral | Number | Absolute at GB | Relative at GB (%) | Spatial pattern | Formula |
|---------|--------|----------------|--------------------|-----------------|---------|
| Sulphates | 386 | 92 | 23.8 | clusters and layers | $XSO_4$ |
| Gypsum | 170 | 47 | 27.6 | throughout | $CaSO_4 * 2H_2O$ |
| Quartz | 126 | 28 | 22.2 | NA | $SiO_2$ |
| Mica | 81 | 22 | 27.2 | limited area | $(K, Na, Ca, NH_4)Al_2(Si_3Al)O_{10}(OH)_2$ |
| Feldspar | 67 | 21 | 31.3 | NA | $(K, Na, Ca, NH_4)(Al/Si)_4O_8$ |
| Mg-sulphate | 64 | - | - | clusters and layers | $MgSO_4$ |
| Bloedite | 61 | - | - | clusters and layers | $Na_2Mg(SO_4)_2 * 4H_2O$ |
| Nitrates | 36 | 9 | 25 | throughout | $XNO_3$ |
| Hematite | 33 | 8 | 24.2 | NA | $Fe_2O_3$ |
| K-sulphate | 31 | - | - | clusters and layers | $K_2SO_4$ |
| Krohnkite | 27 | - | - | clusters and layers | $Na_2Cu(SO_4)_2 * 2H_2O$ |

GB=grain boundaries, NA=no common pattern

### 3.2 Raman spectra and derived mineralogy

We conducted Raman spectroscopy on several of the previously located micro-inclusions from all eleven samples. Analysed inclusions were chosen randomly, but with the intention to represent the entire area of the sample. The use of confocal mode and a 20x and 50x lens allowed us to find more than 97 % of the micro-inclusions indicated in our impurity maps. The number of analysed micro-inclusions per sample ranged from 47 to 134 and a total of 791 spectra were identified. Luminescence was present in 131 measurements, which were excluded. Micro-inclusions with no obtained Raman signal could be Raman-inactive or the Raman signal might be superimposed by the ice signal, i.e. too low in intensity to be identified with confidence. Some analysed micro-inclusions showed Raman spectra of more than one mineral. We classified these complex micro-inclusions (Fig. 2D and Fig. 7B) as one spectra of each type regardless of the relative strength of the Raman peak as done by e.g., Sakurai et al. (2011).

We identified 26 different Raman signals, which can be mainly differentiated into mineral dust and sulphates. An overview is presented in Fig. 4 and the most abundant minerals are displayed in Table 2. Mineral dust minerals are primarily silicates such as quartz and members of the feldspar or mica group. Less abundant dust minerals are hematite, rutile and anatase (both $TiO_2$, rutile is the high-temperature form), titanite ($CaTiSiO_5$), jacobsite ($MnFe_2O_4$). Sulphates can be difficult to distinguish due to tiny differences in their Raman spectra, but most sulphates were identified (Fig. 4B). Most abundant are gypsum, Mg-sulphate, and bloedite. Less abundant are krohnkite, Na-, and K-sulphates. Spectra with a strong peak at 1050 $cm^{-1}$ probably indicate K-nitrates while a strong peak around 1070 $cm^{-1}$ might indicate Na-nitrates (Ohno et al., 2005). The carbonate dolomite ($CaMg(CO_3)_2$) was identified at 1256.96 m in large quantities. We further measured Raman spectra
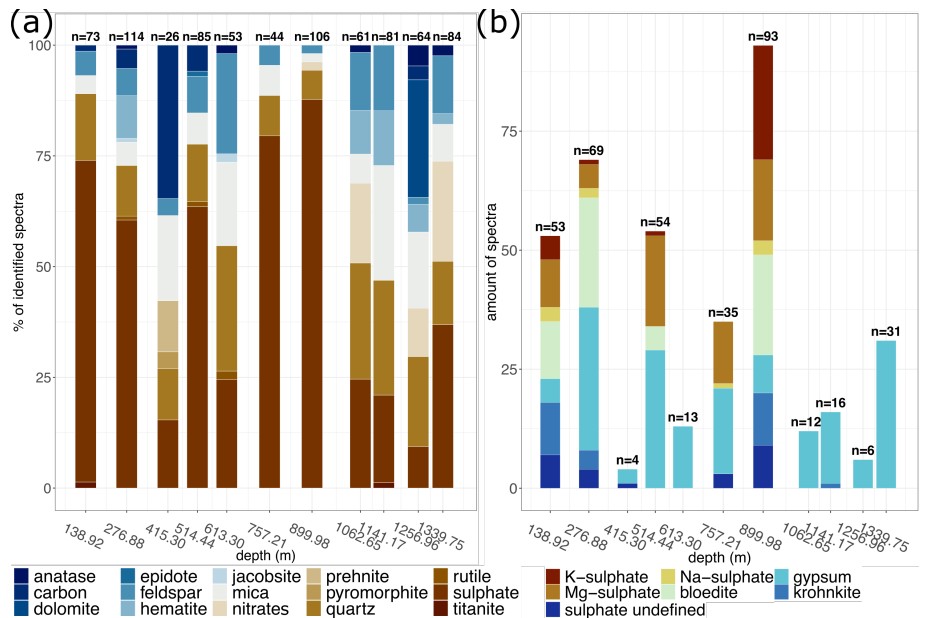

**Figure 4.** Identified Raman spectra of micro-inclusions in EGRIP Holocene ice, n is the total amount of identified spectra per sample. The two deepest samples are from the Bølling Allerød and have by far the highest insoluble particle concentrations. A) All identified Raman spectra per sample. For better visibility some Raman spectra are condensed in groups (e.g., sulphates and mica). B) Identified sulphates in detail. Sulphate diversity decreases below 900 m.

which might be carbonaceous particles (C), pyromorphite ($Pb_5(PO_4)_3Cl$), prehnite ($Ca_2Al_2Si_3O_{10}(OH)_2$), and epidote ($Ca_2(Fe/Al)Al_2(Si_2O_7)(SiO_4)O(OH)$), the latter three were only found once. It is likely that the carbonaceous particles originate from biomass combustion, i.e. black carbon.

### 3.3    Mineral diversity with depth

Between eight and 17 different spectra were identified per sample, the median value is 10 different spectra per sample. To
compare our samples despite the varying amount of total identified Raman spectra per sample we calculated a diversity index $I_{var}$ with the total amount of identified micro-inclusions per sample ($n_i$) and the amount of different minerals per sample ($n_m$):

$$I_{var} = \frac{n_i}{n_m} \tag{1}$$

Mineralogy diversity increases slightly with depth (Fig. 5). Deeper samples are slightly more diverse in mineralogy even
though the large diversity of sulphates is only found in the upper 900 m (Fig. 4B). As expected, samples with larger numbers of identified Raman spectra tend to be more diverse in mineralogy supporting the use of $I_{var}$.

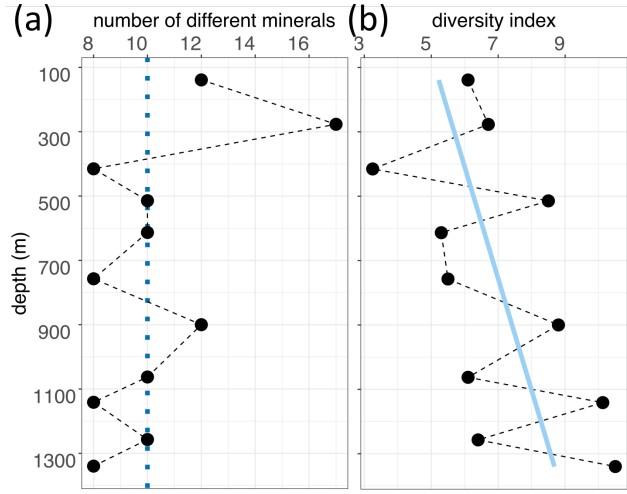

**Figure 5.** Mineral diversity with depth in EGRIP ice. A) Absolute numbers of different minerals per sample. The dotted blue line is the median value (10). B) Diversity index values calculated after Eq. (1). The light blue line is a linear regression. Higher values indicate a larger mineral diversity in relationship to the amount of identified Raman spectra per sample.

## 3.4 Spatial patterns revealed by mineralogy

### 3.4.1 Large-scale: Mineralogy evolution along the ice core

The relative abundance of the three silicates quartz, feldspar and mica often correlates, with increased values at the same depths
(e.g., at 276.88 m, 613.3 m, 1062.65 m). Quartz represents between 6.6 % (899.94 m) and 28.3 % (613.3 m), feldspar between 1.6 % (1256.98 m) and 22.6 % (613.3 m), and mica between 1.9 % (899.94 m) and 25.9 % (1141.2 m) of all identified micro-inclusions at one depth. These minerals dominate the sample from 613.3 m of depth and the three samples between 900 and 1257 m. The deepest sample shows an additional high amount of sulphates, i.e. gypsum (Fig. 4). Minerals, such as hematite, titanite or nitrates, were only found in the samples below 900 m. Nitrates were found at 899.98 m and were among the main
impurities in three of the four deepest samples, i.e. at 1062.65, 1256.96 and 1339.75 m of depth. Dolomite was only identified at 1256.96 m of depth. Carbonaceous particles were found in the four shallowest samples (138.92, 276.88, 415.3, 514.44 m) and at a depth of 1256.96 m.

The majority of sulphates was found in the seven shallowest samples below a depth of 900 m (Fig. 4B). Over the core sulphate spectra represent between 9.4 % (1256.98 m) and 87.7 % (899.94 m) of all identified micro-inclusions at one depth.
As displayed in Fig. 4B and briefly described in Sect. 3.3 various different sulphates are found above 900 m while the only sulphates below 900 m are gypsum and, at 1256.96 m, one krohnkite micro-inclusion. Gypsum sometimes follows the abundance pattern of the terrestrial dust minerals (at 276.88 m, 514.44 m, 1141.17 m).





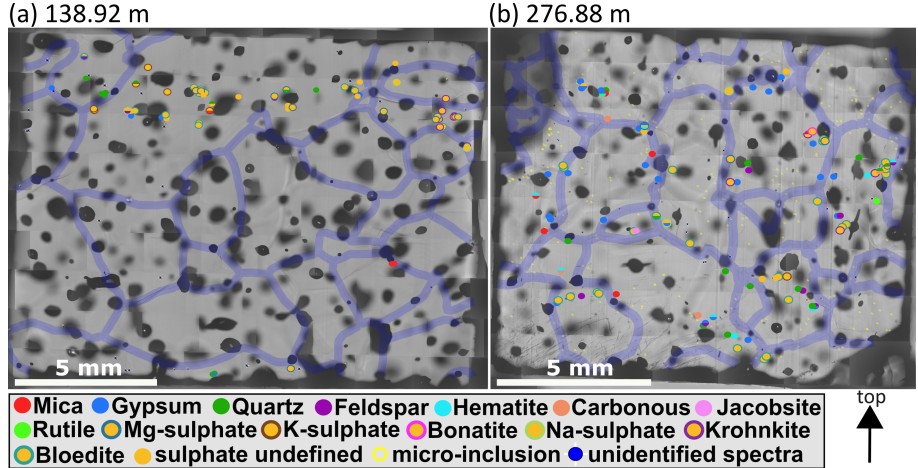

**Figure 6.** Impurity maps from a depth of 138.92 and 276.88 m. Identified micro-inclusions are represented by filled circles with different colours; sulphates are orange circles with differently coloured contours. Transparent violet lines are 300 $\mu$m thick grain boundaries, the arrow points towards the surface of the ice sheet. A) A layer of sulphate micro-inclusions is in the upper part of the sample, below are almost no micro-inclusions. B) A wide variety of different Raman spectra and spatial patterns, e.g, clusters and rows of sulphates.

### 3.4.2 Small-scale: Clusters

Dense clusters of micro-inclusions (minimum of three proximal inclusions) of the same mineralogy were observed throughout
the core, but especially above 900 m (e.g., at 138.92, 514.44, 757.21, and 899.98 m). Specifically sulphates at these depths were often found in such clusters (Fig. 6 and 7A, B). The abundance of such clusters of micro-inclusions with similar chemistry declines with depth and is rarely observed below 900 m. Certain minerals, such as mica, only occur in a limited area of the sample (e.g., at 1339.75 m) without creating distinct layers or clusters. Gypsum and nitrates were found throughout the entire sample area while no specific localisation was identified for minerals such as quartz, hematite or nitrates (Table 2).

### 3.4.3 Small-scale: Horizontal layers

Often micro-inclusions of the same mineralogy are found in distinct layers horizontal to the core axis (distinct vertical layers were not observed) or broader bands. Fig. 7A shows that at a depth of 757.2 m mineralogy changes on the centimetre-scale. The shallowest part of the sample is characterised by a mixture of sulphates (mainly gypsum) and terrestrial dust (quartz, feldspar, mica). Some centimetres deeper a small layer of gypsum is followed by a layer of mainly Mg-sulphates. At the bottom of the
sample Mg-sulphates dominate, accompanied by gypsum and mica.

A broad layer is observed at 899.98 m (Fig. 7B). The upper three quarters of the sample are dominated by different sulphates, especially the middle section shows a clear layer. Sulphates in this layer are often found in clusters. The deepest part of the sample inhibits a broader variety of minerals, ranging from different sulphates to mica and quartz (Fig. 7B).





### 3.4.4 Small-scale: Localisation at grain boundaries

181 identified micro-inclusions were located at grain boundaries, i.e. in the upper-limit assumption of 300 $\mu$m thick grain boundaries. 92 sulphate particles (including 47 gypsum inclusions) were found at grain boundaries, followed by quartz (28), mica (22), feldspar (21), nitrates (9), hematite (8), and anatase and titanite (both 1). In relation to the total number of each mineral 31.3 % of feldspar, 27.6 % of gypsum, 27.2 % of mica, 23.8 % of sulphate (including gypsum, 20.8 % without gypsum), and 22.2 % of quartz was located at grain boundaries.

The relative amount of sulphates at grain boundaries was higher at depths with a high diversity in sulphate types, i.e. at 138.92, 276.88, 514.44, 613.3, 757.21, and 899.98 m, than the relative amount of terrestrial dust located at grain boundaries. Below 900 m feldspar, mica and quartz were more common at grain boundaries than sulphates, i.e. gypsum. However, at 1339.75 m 37.9 % of all identified micro-inclusions at grain boundaries were gypsum.

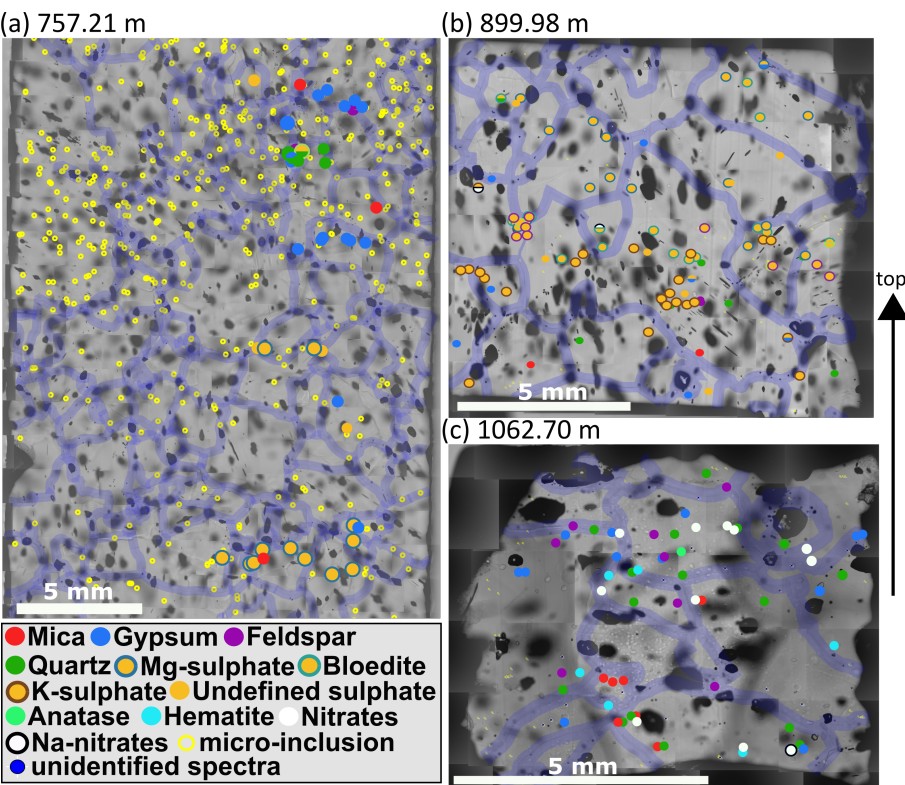

**Figure 7.** Impurity maps from a depth of 757.21, 899.98, and 1062.7 m. Same annotations as in Fig. 6. A) Mineralogy changes within centimetres along the vertical axis. The shallower part is dominated by gypsum and terrestrial dust while the deeper part is dominated by Mg-sulphates. B) A strong sulphate layer is found in the middle of the sample, terrestrial dust is mainly found below. C) Mineralogy differs strongly from the shallower samples: sulphates are less diverse, mineral dust and nitrates dominate, and clusters are less common.



## 4 Discussion

In this study we have shown that the insoluble particle content is variable on the centimetre-scale, but increases with depth. Furthermore, the mineralogy of micro-inclusions in EGRIP ice varies on the large-scale, i.e. throughout the core, and on small spatial scales, i.e. centimetres. The accompanying data from the applied methods (CFA, microstructure-mapping and Raman spectroscopy) enabled us to create high-resolution impurity maps showing that sulphates and terrestrial dust are the most abundant minerals. The upper 900 m are characterised by sulphates with a diverse mineralogy while mineral dust minerals and 270 gypsum dominate below. Spatial patterns of inclusions of the same mineralogy were revealed, ranging from distinct layers of several millimetre thickness to clusters (mainly of sulphates).

### 4.1 Insoluble particles at EGRIP

We here present the 55 cm bag mean CFA results of the insoluble particle number (Fig. 1), representing the dust concentration in the ice. Though, detailed studies will follow in the future, we can already see, that the insoluble micro-particle concentration, 275 interpreted as the mineral dust aerosol particle record, seems to vary through the Holocene. The concentration is comparable to e.g., the North Greenland Ice Core Project ice core (Steffensen et al., 2008) and is low in overall concentration compared to Glacial ice (e.g., Steffensen, 1997; Ruth et al., 2003).

High concentrations of insoluble particles measured with CFA correlate with areas of numerous micro-inclusions observed with optical microscopy and Raman spectroscopy. The samples with the highest insoluble particle counts, i.e. at depths of 280 1256.96 and 1339.75 m, are from the Bølling Allerød. This explains the much higher insoluble particle concentration, compared to our Holocene samples (Fig. 1), which is closer to resemble concentrations found in Glacial ice (Ruth et al., 2003). The highest amount of insoluble particles measured with CFA is at 1256.96 m of depth (Fig. 1K), inside a cloudy band. Cloudy bands originate from seasonal events, such as atmospheric storms, resulting in increased transport and deposition of impurities on the ice sheet (Svensson et al., 2005). This explains the high number of mineral dust minerals, the low number of sulphates 285 measured with Raman spectroscopy (Fig. 4A) and the resulting lower $I_{var}$ compared to the deepest sample (1339.75 m). It further supports the hypothesis that insoluble dust contributes significantly to cloudy bands (Svensson et al., 2005). The unique location of the EGRIP drill site on an ice stream and the assumed high impact of high-impurity layers on the deformation of the ice (e.g., Dahl-Jensen and Gundestrup, 1987; Paterson, 1991; Miyamoto et al., 1999) highlight the need for an in-depth study of cloudy bands at EGRIP.

We adjusted the approach of using CFA data for efficient Raman analysis (Eichler et al., 2019) and conclude that analysing insoluble particle data first on the large-scale, i.e. along one ice core (Fig. 1A), followed by small-scale analysis, i.e. over centimetres (Fig. 1B-L), provides an excellent basis for a systematic analysis of different depth regimes while ensuring a sufficient number of micro-inclusions. Combined with microstructure-mapping it additionally accelerates the time-consuming search for micro-inclusions, especially in pure Holocene ice.


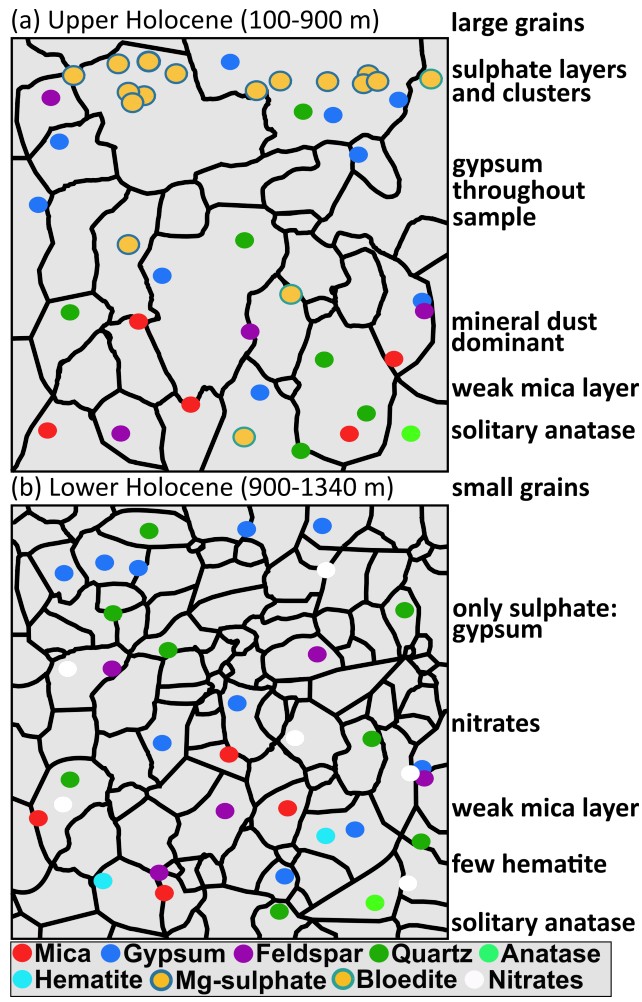

**Figure 8.** Sketch of the simplified distributional patterns of minerals found in A) upper Holocene ice and B) lower Holocene ice.

## 295  4.2  Abundance of minerals in EGRIP ice

In general, our results show a large abundance of minerals across all samples while depth-related differences occur in total, and relative, mineral abundance. A detailed discussion of the observed shift in mineralogy over the EGRIP ice core is presented in Sect. 4.2.1.

The most abundant minerals are sulphates and terrestrial dust minerals, which agrees with observations by e.g., Ohno et al.
(2005); Sakurai et al. (2009); Eichler et al. (2019). Antarctic ice analysed by Ohno et al. (2005, 2006); Sakurai et al. (2010, 2011) and Eichler et al. (2019) is dominated by sulphates, they represent e.g., up to 96 % of all identified spectra in the interglacial samples of Eichler et al. (2019). Contrary to this, our samples show, in general, a higher diversity in minerals as displayed in Fig. 5. Only the sample from 899.98 m of depth has a comparable amount of sulphates (87.7 % of all identified





spectra). We identified most sulphates as gypsum or Na-sulphates, which agrees with e.g., Ohno et al. (2005); Eichler et al.
(2019) (Fig. 4B).

We observed a total of 170 gypsum micro-inclusions making it the most abundant mineral and the only sulphate which occurs in every sample supporting e.g., Legrand and Mayewski (1997) and Sakurai et al. (2009). Legrand and Mayewski (1997) show that $CaSO_4$ and $MgSO4$ can be produced if terrestrial compounds, such as $MgCO_3$ and $CaCO_3$, are neutralised with $H_2SO_4$. They also found $CaSO_4$ in Holocene ice from Greenland despite a low $Ca^{2+}$ concentration.

Mineral dust, such as feldspar, mica and quartz, were abundant at all depths, but generally in lower abundances than gypsum (Table 2). These minerals were especially abundant at a depth of 613.3 m and in the deepest sample close to the Glacial. The Glacial sample analysed by Eichler et al. (2019) shows a similar abundance of terrestrial dust minerals.

We observed nitrates at depths between 899.94 and 1339.75 m. They were also found by Fukazawa et al. (1998); Ohno et al. (2005), but not with Raman spectroscopy by Sakurai et al. (2009) and Eichler et al. (2019) even though these authors observed
nitrate-ions with IC. Nitrates are a major impurity component in polar ice as obtained from CFA (Röthlisberger et al., 2000b) and IC (Eichler et al., 2019) analyses, but there is a lack of understanding in which form they are present in ice. For example, sulfuric acid competes with other acids to react with the relative rare cations, replaces other acids in their salts and thus forms a variety of sulphate salts (Iizuka et al., 2008). The relative abundance of nitrates at a certain depth (Fig. 4A) indicates that similar processes occur with nitrates. However, nitrate ions seem to be more likely to exist in dissolved forms than in particle
forms as suggested by Eichler et al. (2019).

### 4.2.1 Shift in mineralogy

We have observed a pronounced shift in mineralogy from a sulphate and terrestrial dust-dominated regime in the upper 900 m to a terrestrial dust-dominated regime with partially high amounts of gypsum. Similar findings were reported by earlier studies (e.g., Ohno et al., 2005; Sakurai et al., 2009; Eichler et al., 2019) who found sulphates and mineral dust in varying numbers,
but our systematic approach exposes the mineral diversity already present in Holocene ice in more detail.

Sulphates and terrestrial dust minerals are found throughout the entire ice core, but in varying abundance (Fig. 4 and simplified in Fig. 8). Rare minerals, such as hematite, anatase and titanite were found at all depths. Other minerals e.g., rutile and epidote, were only found in shallow samples while e.g., dolomite and nitrates were only found in deep samples. The high abundance of sulphates in many samples can be explained by the formation of sulphates after initial deposition (Ohno et al.,
2005). $Na^+$ and $Mg^{2+}$, the main cations combining with $SO_4^{2-}$, could originate from blown-in sea salt ($NaCl$ and $MgCl_2$), which was deposited on the ice sheet and, during transport, has partially chemically reacted with acidic compounds as proposed by Artaxo et al. (1992); Kerminen et al. (2000) for Antarctic sea salt. However, deeper samples show smaller amounts, and varieties, of sulphates, which almost completely disappear below 900 m. In the four deepest samples gypsum is the only sulphate, except for one Krohnkite micro-inclusion at 1141.17 m. Data is scarce, but e.g., Sakurai et al. (2009, 2011) and Eichler
et al. (2019) show that sulphates are found in deeper parts of ice sheets, also consisting almost entirely of gypsum supporting our results.



Even though we found a shift in mineralogy, this shift does not correlate with a change of the climate period, i.e. from Glacial to Holocene ice. Eichler et al. (2019) found a strong difference in mineralogy between Interglacial and Glacial ice in the EDML ice core. However, main minerals in Interglacial ice are sulphates (especially Na-sulphate and gypsum) while
mineral dust and gypsum dominate in Glacial ice (Eichler et al., 2019). Ohno et al. (2005) show that salt inclusions consist mainly of Na- and Mg-sulphates in Holocene ice from Dome Fuji, Antarctica while gypsum and nitrate salts dominate in the Last Glacial Maximum. This agrees with our findings and could indicate that, in Greenland and Antarctica, sulphates are more abundant in warm periods, such as the Holocene or the Eemian, while mineral dust and gypsum dominate during cold periods, such as the last Glacial. However, it is interesting that the transition from the Holocene to the Stadial is not represented by a
major change in mineralogy. A closer investigation of EGRIP glacial samples will show if prominent changes in mineralogy also occur in deeper depth than the analysed upper 1340 m.

### 4.2.2  Detailed comparison with mineralogy in the GRIP ice core

A limited number of polar ice cores have been drilled over the last decades and mineralogy data is only available for a few of those. The relatively close to EGRIP located drill site of the GRIP ice core was analysed with Raman spectroscopy and EDS
by Sakurai et al. (2009). We thus compare both cores here in detail.

We found gypsum at all depths at EGRIP resembling the results by Sakurai et al. (2009). Similar to our findings, Na- and Mg-sulphates, quartz, and nitrates are less abundant in their samples. Sakurai et al. (2009) analysed 6.9 and 9.8 ka BP old samples, which we compare to our 6.4 ka, 9.3 ka and 10.2 ka old samples. The younger samples from both studies are mainly composed of gypsum, Na- and Mg-sulphates, and smaller amounts of quartz, feldspar, and mica. In the 9.8 ka old sample
Sakurai et al. (2009) found gypsum, slightly less Na- or Mg-sulphate and some "other" minerals while our 9.3 ka and 10.2 ka old samples show a slightly different mineralogy. Quartz is dominant in both samples and is accompanied by gypsum, feldspar and, to varying degrees, mica. Both samples contain no sulphates except gypsum and krohnkite. Gypsum is the only sulphate in our oldest sample (14.1 ka), which agrees well with the 13.5 ka sample from Sakurai et al. (2009).

The higher amount of terrestrial minerals in our samples is difficult to explain conclusively, because the GRIP and EGRIP
sites are comparably close to each other and in similar distances to the coast. At certain depths, e.g., at 613.3, 1062.65, and 1256.96 m, the majority of micro-inclusions consists of mineral dust, which might indicate strong dust storms or deposition events as proposed for cloudy bands by e.g., Svensson et al. (2005). However, we show that the diversity in mineralogy can be high within centimetres (e.g., Fig. 7). The observed differences are thus probably caused by natural variability amplified by methodological differences, such as different sample sizes.

### 4.2.3  Methodological drawbacks

We can not determine if rarely found minerals only occur at certain depths, or if this is related to statistics drawbacks. Minerals, such as hematite and anastase, were only found in a few samples while rutile, titanite, epidote, and jacobsite have only been found once. Unfortunately, the problem of statistics reoccurs in microstructural impurity research, which can only be partly solved by analysing larger samples, and with a higher spatial resolution.



Furthermore, there are known drawbacks of Raman spectroscopy, such as the inability to identify Raman-inactive compounds, the superimposition of the ice spectra, and the focus on visible, undissolved impurities. Especially the main component of sea salt, NaCl, is inaccessible with Raman spectroscopy and thus not considered in our analysis.

    Some Raman spectra from a depth of 415.3 m have been difficult to identify. We thus had to exclude several spectra, further lowering the already small number of observations. However, analysing inclusions inside the ice prevents contamination and
preserves their in situ locations and is thus a well justified approach.

    We could not identify all measured Raman spectra similar to other studies using this method. As an example we measured the same unknown spectra as presented in Fig. 2G by Sakurai et al. (2011). Obtaining good Raman spectra and identifying them with enough confidence will always be a challenge. These technical limitations hamper a fully holistic analysis and are difficult to overcome. A partial solution would be an analysis of the same samples with a subsidiary method, such as SEM
coupled with EDS or LA-ICP-MS. Such a combined analysis would enable a deeper insight into the EGRIP ice core and into impurities in polar ice in general.

### 4.3   Possible reasons for the diversity of minerals in EGRIP ice

We observed a prominent diversity in mineralogy of micro-inclusions in EGRIP ice across all scales. Mineral abundance changes throughout the core, i.e. the observed shift in sulphate diversity (Sect. 4.2.1), and across samples, i.e. in distinct layers
and clusters (Sect. 3.4). This can partly be explained methodologically, e.g., by the number of analysed micro-inclusions per sample, by different deposition conditions, or by chemical reactions taking place in the ice.

    A difference in the chemistry at the ice sheet surface at the time of deposition and different atmospheric circulation patterns, and thus varying aerosol input, could be indicated by our CFA data (Fig. 1). Iizuka et al. (2012) concluded that aerosol sulfatisation in Antarctica is more likely to occur in the atmosphere or during fallout than after deposition. For Greenland, the major
dust source areas for the Last Glacial and the Younger Dryas were the East Asian deserts (Svensson et al., 2000; Vallelonga and Svensson, 2014) with minor contributions from the Sahara (Han et al., 2018). Aerosols deposited in the Holocene are likely from the Takla Makan and Gobi desert (Bory et al., 2003; Vallelonga and Svensson, 2014). These differences in dust input between the Holocene and the Last Glacial are represented in our results, e.g., the highest $I_{var}$ value was measured in the deepest sample originating from the Bølling Allerød. The strong layering and clustering of sulphates could originate from dry
deposition events, which form deposition crusts mainly containing sulphates. Another possible aspect is the unique impact of NEGIS. Older samples originate from further upstream than younger samples, which could be reflected in the micro-inclusion mineralogy. However, we do not expect any major upstream deposition effects during the Holocene, but this should be verified in future studies. A systematic follow-up study on EGRIP Glacial ice is needed to investigate if the observed trends, e.g., of mineral diversity and clustering, continue with depth.

Another possibility explaining the diversity in chemical compounds are chemical reactions occurring inside the ice as summarised for snow by Bartels-Rausch et al. (2014), and discussed for ice by Steffensen (1997); de Angelis et al. (2013); Baccolo et al. (2018) and Eichler et al. (2019). Time spans of several thousand years could enable that impurities in the ice react and thus, led to post-depositional changes in the composition of impurities. Barnes et al. (2003b); Masson-Delmotte et al. (2010);





de Angelis et al. (2013) suggest movement of chloride and sulphate ions after deposition while Barnes and Wolff (2004) re-
port of possible reactions between dust particles and soluble impurities. Baccolo et al. (2018) suggest that, for iron, oxidation
and dissolution are the main processes taking place in deep samples from the Talos Dome, Antarctica ice core. The deep ice
sheet environment is not anoxic and dissolved oxygen and liquid water veins might support the oxidation and dissolution of
specific mineral phases (Baccolo et al., 2018). Fe-minerals occur relatively often in polar ice and were identified in this study
and by e.g., Baccolo et al. (2018); Eichler et al. (2019); Fe was found by e.g., Obbard and Baker (2007); Della Lunga et al.
(2014). Furthermore, Faria et al. (2010) observed the formation of solid inclusions in deep ice, which was supported by Eichler
et al. (2019). A local mixing of impurities in shear bands with high strain rate and strain could explain the observed cluster-
ing of sulphates (Eichler et al., 2019). There might be local small-scale processes involved leading to preferred clustering of
micro-inclusions with similar chemistry. We observed preferred clustering at all depths, however samples below 900 m show
significantly fewer clusters. This correlates with the depth of declining sulphate-diversity and could indicate that, around this
depth, certain unknown chemical reactions occur or that large-scale boundary conditions, such as climate or ice sheet extent,
changed during the time of deposition.

## 4.4   Holocene climate as derived from ice cores and its possible imprint in mineralogy

Previous studies have suggested two possible scenarios for the Holocene climate evolution. It was either a a stable period of
climate with the Holocene Thermal Maximum in the Mid-Holocene (9-5 ka BP), or a period with a rather long-term cooling
trend and an earlier, stronger Holocene Thermal Maximum (Axford et al., 2021). The prominent change of abundant minerals
at a depth of 900 m could be explained by these two scenarios leaving different imprints on the mineralogy.

The extension of the Greenland Ice Sheet during the Holocene cannot be constrained exactly, and it probably differed
significantly between regions (Lecavalier et al., 2014; Young and Briner, 2015). The rapid retreat of the ice sheet in NE-
Greenland 12 ka ago lead to ice-free coastal areas in NE-Greenland after ~9 ka BP (Lecavalier et al., 2014). Air temperatures
over Greenland increased in the Holocene Thermal Maximum and peaked ~7.8 ka BP (Lecavalier et al., 2014; Axford et al.,
2021) exposing bedrock and thus enabling local dust sources. Syring et al. (2020) show that the sea ice cover in NE-Greenland
transformed from a reduced state towards a marginal, almost extended, state between Early (~11.7-9 ka BP) and Mid Holocene
(~9-5 ka BP) possibly impacting the input of sea salt, a major source of sulphates. Even though the main dust input to Greenland
was from East Asia (Svensson et al., 2000; Bory et al., 2003), regional changes (e.g., exposed bedrock) might have had an
impact on the availability of chemical compounds in the air, and thus during deposition. However, this is unlikely due to the
high elevation of EGRIP (2700 m) and the upstream effect caused by NEGIS, but detailed isotopic and mineralogical analyses
are needed to test this.

The occurrence of other sulphates than gypsum in 9.8 ka old GRIP ice (Sakurai et al., 2009) contradict a general atmospheric
signal for Greenland and support a site specific difference at EGRIP. A major change in aerosol chemistry and ion composition
deposited at EGRIP is however unlikely, but upcoming studies are needed to discuss the chemical evolution at the site. The
average EGRIP annual layer thickness is surprisingly constant during the Holocene and a significant change occurs around 7-8
ka b2k (Mojtabavi et al., 2020), but the involved upstream effect of NEGIS and the general history of the ice stream are not well





understood, thus hampering the further investigation of these, spatial and temporal, large-scale processes. Visual Stratigraphy and the core break record show that the brittle zone, the transition from air bubbles to clathrates hyrates (Ohno et al., 2004; Neff, 2014), at EGRIP is roughly at 550-1000 m of depth. Defects in the ice matrix, grain boundaries and micro-inclusions can act as nucleation sites for hydrate nucleations (Ohno et al., 2010). We found most clathrates below 900 m evoking the possibility that this transition zone also impacts sulphate chemistry, but details are unknown.

This brief evaluation of possibilities shows that chemical reactions and differences in the salt-formation conditions in the ice are the most likely explanation, as e.g., suggested for differences in micro-inclusion chemistry between Dome Fuji Holocene and Interstadial ice (Ohno et al., 2005). It is thus necessary to investigate the redistribution of impurities, and their chemical reactions (including their rates), in ice sheets. Samples with a high number of identified Raman spectra mostly showed a higher diversity in mineralogy, which leads to the conclusion that more extensive impurity studies are needed to get comprehensive results. At this point, we can only suggest explanations for the change in diversity of sulphates, but upcoming studies on e.g., the bulk chemistry of EGRIP Holocene ice and flow effects of NEGIS might deliver crucial information to fully explain it.

## 4.5 Outlook

The data presented in this study and its companion paper show the need to aim for a more holistic and in-depth understanding of the, macro- and micro-scale, processes in polar ice. Progressing towards an improved synoptic view is only possible by broader collaborations between the involved diverse research fields (microstructure, paleo-processes, impurities) applying different methods. Therefore, it is needed to clearly define method-related limitations, to develop concepts to tackle these limitations while systematically combining their strengths. Especially the broad field of cryo-related impurity research has a variety of methods to measure dissolved and undissolved impurities, but they are mainly applied solitarily and results are often only partly transferable. To illustrate this point we briefly summarise the (simplified) capabilities of established methods regarding their ability to measure dissolved or undissolved impurities. IC measures dissolved (and a fraction of undissolved) impurities while Raman spectroscopy measures a limited number of undissolved particles, such as the micro-inclusions analysed in this study. CFA melts the ice, where an unknown fraction of insoluble particles in the ice go into solution. The remaining insoluble particles are measured by the particle sensor of the CFA, giving a relative number of particle concentration of the original ice. Finally, ICP-MS and LA-ICP-MS measure all, dissolved and undissolved, particles but information on the fraction of both impurity states are lost. Recent work by Ng (2021) demonstrates the challenge up ahead of better understanding climate signal records in deep polar ice and how these signals could be affected by processes involving grain boundaries, microstructure, veins, and impurities.

Our presented data and drawn conclusions demonstrate that our systematic approach is a good way towards a more detailed understanding of impurities, i.e. their mineralogy and location, in polar ice. It emphasises the importance of systematically analysing different spatial scales with a variety of methods to achieve holistic results. Despite the comparably high absolute number of identified micro-inclusions statistics are still improvable, especially for minerals which occur rarely such as hematite or anatase. Further in-depth analysis of EGRIP Glacial ice could help to improve statistical certainty. A comprehensive un-



derstanding of the mineralogy and location of impurities in our samples could be reached by analysing our samples with a complimentary method, such as LA-ICP-MS or SEM and EDS.

## 5    Conclusions

We here derive the first systematic analysis of the mineralogy of micro-inclusions in Holocene and Late Glacial ice from the
EGRIP ice core, the first deep ice core from a fast flowing ice stream. More specific, we derived dust particle concentration along the upper 1340 m via CFA and analysed micro-inclusions in detail with a combination of optical microscopy and Cryo-Raman spectroscopy at eleven different depths.

By identifying almost 800 Raman spectra we obtained new qualitative, and quantitative, insights into the mineralogy of micro-inclusions in polar ice. 26 different spectra were identified, which indicates a more diverse mineralogy in ice from the
last 14 ka than previously observed. In general most inclusions are sulphates, especially gypsum, and mineral dust, such as quartz, mica, and feldspar. Sulphates in the upper 900 m tend to create clusters of similar chemical composition and mineral diversity increases slightly with depth. The here analysed 1340 m of the EGRIP ice core can be divided into two depth regimes of different mineralogy. The upper 900 m are characterised by various sulphates while below this depth gypsum is the only sulphate. This might be explained by different, yet uncharted, chemical reactions occurring in the ice or by large-scale changes
in NE-Greenland during the Mid-Holocene, such as the decreasing lateral extent of the Greenland Ice Sheet revealing bedrock at the coast or the varying sea-ice cover impacting the availability of sulphates in the air during deposition.

Our study emphasises the need to overcome technical limitations, which could be achieved by an inter-method comparison of e.g., Raman spectroscopy, LA-ICP-MS and SEM. However, our systematic overview of the mineralogy of micro-inclusions throughout a large part of one deep ice core helps to develop a better understanding of the role of impurities in polar ice.

*Data availability.*   High-resolution microstructure and impurity maps were submitted to PANGAEA and will be available eventually. Raman data are available on request. CFA data will be published separately with an in-depth CFA study.

*Author contributions.*   Conceptualization by NS, IW, and MH. Microstructure mapping and Raman methodology developed by NS, JE, and IW, CFA methodology by TE and CJ. Investigation and Data curation by NS, TE, and CJ. Formal analysis by NS and TE. Funding acquisition by IW. The original draft was written by NS with assistance from all co-authors.

*Competing interests.*   The authors declare that the research was conducted in the absence of any commercial or financial relationships that could be construed as a potential conflict of interest.



*Acknowledgements.* This work was carried out as part of the Helmholtz Junior Research group "The effect of deformation mechanisms for ice sheet dynamics" (VH-NG-802). We thank all EGRIP participants for logistical support, ice processing and fruitful discussions. EGRIP is directed and organised by the Centre for Ice and Climate at the Niels Bohr Institute, University of Copenhagen. It is supported by funding

agencies and institutions in Denmark (A. P. Møller Foundation, University of Copenhagen), USA (US National Science Foundation, Office of Polar Programs), Germany (Alfred Wegener Institute, Helmholtz Centre for Polar and Marine Research), Japan (National Institute of Polar Research and Arctic Challenge for Sustainability), Norway (University of Bergen and Trond Mohn Foundation), Switzerland (Swiss National Science Foundation), France (French Polar Institute Paul-Emile Victor, Institute for Geosciences and Environmental research), Canada (University of Manitoba) and China (Chinese Academy of Sciences and Beijing Normal University). TE and CMJ gratefully ac-

knowledge the long-term financial support of ice core research at the University of Bern by the Swiss National Science Foundation (grant no. 200020_172506 (iCEP) and 20FI21_164190 (EGRIP)).



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
