# Peer review of "Microstructure, Micro-inclusions and Mineralogy along the EGRIP ice core - Part 2: Implications for paleo-mineralogy"

_The Cryosphere, 2021_

## Author Response (AR1)

**Replies to referees:**
**Referee 1**
Line numbers indicate the new locations of changes applied to the revised manuscript.

**Ref1:** It was my pleasure to review "Microstructure, Micro-inclusions and Mineralogy along the EGRIP ice core - Part 2: Implications for paleo-mineralogy". This paper presents a novel dataset of impurities in a Greenland ice core. The data is interesting, and the methods properly presented. However, the introduction and discussion could be considerably improved.

The introduction brings in a great deal of previous research but doesn't make clear points about the knowledge gap that the paper is addressing. Nor does it state any hypotheses or provide much theoretical framework on which hypotheses could be made. The importance of impurities is brought up and the word important is used repeatedly. But nowhere is the link made between this particular research effort and its applications to ice mechanics, etc. Neither is the later discussions of chemistry and mineralogy at all prefigured in the introduction.

**Reply:** We thank the referee and value the input and ideas. We have gone through all comments and adopted most of them. Answers to the specific comments can be found below.

We changed the introduction to clarify knowledge gaps and to define our objectives more clearly. This will help to prefigure the discussion and goes as follows (**l. 62**): "…The impurity content plays a major role regarding the physical properties of snow, firn, and ice, such as electrical conductivity (Alley and Woods, 1996; Wolff et al., 1997), permittivity (Wilhelms et al., 1998), and mechanical properties (e.g., creep behaviour, dislocation velocity) (e.g., Jones and Glen, 1969; Dahl-Jensen and Gundestrup, 1987; Paterson, 1991; Hörhold et al.,2012). Several studies investigated the impact of impurities on the deformation of ice (e.g., Jones and Glen, 1969; Petit et al., 1987; Iliescu and Baker, 2008; Eichler et al., 2019), which in turn influences the flow of ice - a major uncertainty regarding future projections of ice sheet behaviour and solid ice discharge. This chain of processes at the micro-scale resulting eventually in the large-scale deformation behaviour of ice needs to be better understood, especially in fast flowing ice, as present in ice streams. The microstructure of ice can be impacted directly by processes related to impurities such as Zener pinning (Smith,1948; Humphreys and Hatherly, 2004) or grain boundary drag (Alley et al., 1986, 1989). These result in changes of the energy, shape and mobility of grain boundaries. Furthermore, impurities such as micro-inclusions directly impact deformation when they form obstacles in the ice matrix and thus enhance strain localisation and the introduction of protonic defects (Glen, 1968) or the formation of dislocation lines (Weertman and Weertman, 1992). In turn, these small-scale processes can also influence climate proxies by altering the stratigraphic integrity of impurity records (Faria et al., 2010; Ng, 2021). The mineralogy of impurities, their localisation and the possible interplay between both certainly plays a role in the interaction of impurities and ice physical properties (e.g., Jones and Glen, 1969; Paterson, 1991) and ice dynamics. Reasons for the current lack in understanding of the whereabouts of impurities in polar ice are manifold, the methodology to systematically look in parallel at localisation (as a microstructural feature) and mineralogy (as a chemical feature) of micro-inclusions has just recently been developed (Eichler et al., 2019). Commonly used methods, developed to derive (paleo-) climate information on impurities such as CFA (e.g., Kaufmann et al., 2008), apply their analysis on melted sample water for the sake of decontamination and time constraints. Doing so, information on the phase and localisation of impurities are not considered. On the other hand, dedicated studies on localisation are spatially very limited and do not catch the overall chemistry of a certain sample. The mineralogy and its possible interplay for localisation was therefore not possible to investigate. However, in order to develop a profound understanding on the mineralogy and localisation of impurities in polar ice, a combination of both approaches is needed.

With our study we attempt to push forward to combine the information of both, microstructure and chemistry to elaborate on the mineralogy of micro-inclusions in polar ice, its possible sources and its interplay with microstructure. There is no investigation of the mineralogy and location of inclusions in one polar ice core with a high enough spatial resolution and reliable statistics to generalise the spatial distribution of inclusions or to investigate the effect of different minerals on ice mechanics. We aim to gain a better understanding of the mineralogical variability of micro-inclusions inside one ice core on the large (hundreds of metres) and on the small (millimetre-centimetre) scale. Was the mineralogy of the deposited impurities stable over the last 14 ka or are there changes following the evolution of the Greenland Ice Sheet? Does the mineralogy of inclusions enable a better understanding of the aerosol content of the past? Furthermore, we investigate if the location of the inclusion in the microstructure is linked to its mineralogy. Our approach is a combination of methods from microstructure and impurity research to perform a systematic high-resolution analysis along one deep ice core. The East Greenland Ice Core Project (EGRIP) ice core, the first ice core drilled through the Northeast

Greenland Ice Stream (NEGIS), enables such an interdisciplinary undertaking. Stoll et al. (2021a) investigate the location of micro-inclusions and their role regarding deformation and microstructure. In this study we investigate the mineralogy of these visible micro-inclusions. We apply bulk insoluble particle measurements from CFA and optical microscopy and Raman spectroscopy on eleven samples along the EGRIP ice core. With this combined data set covering the Holocene, the Younger Dryas, and the Bølling Allerød, i.e. the upper 1340 m, we aim to give a systematic overview of the evolution of the mineralogy of the EGRIP ice core by identifying a relevant number of micro-inclusions at equal depth intervals. We discuss the interpretation of the mineralogy of micro-inclusions in the EGRIP ice core with respect to environmental boundary conditions and survey implications for future research. Possible formation pathways of the observed minerals are briefly discussed, but are not the main focus of this study."

We further changed the last sentence of the abstract to **(l. 15)**: "Our results show that by applying new methods the mineralogy in ice cores, its complexity and importance for localisation studies, opens new avenues for understanding the role of impurities in ice cores."

**Ref1:** Without any research questions initially laid out, the discussion wallows in vague competing hypotheses without definitively defending any point of view. A great deal of the text is dedicated to hypothetical future research, methodological shortcomings, and irrelevant statements about the general state of the science. The text lacks depth of analysis in chemistry or mineralogy. Particularly, the authors need to defend a theory of how the minerals they analysed formed. The questions of whether the sulfate and nitrate minerals are atmospherically formed or formed in ice and snow is not even properly raised, much less resolved. The composition of dissolved ions should be known from the continuous flow analysis, but it isn't included.
**Reply:** We understand, that the focus of the study became not clear and thus changed the introduction (see above). We clarify, that we use the dust particle concentration generated by CFA, but not CFA data (which are not the focus of this study and will be made available eventually by future studies). This is now changed throughout the text in similar ways to the abstract: **(l. 5)** "We use dust particle concentration, optical microscopy and Cryo-Raman spectroscopy to…"

We here combine different approaches from different fields, which makes it difficult to discuss all aspects in depth. We appreciate the comment and changed the text at several places to deliver more depth regarding e.g., formation of minerals. These changes are shown in detail below.
Unfortunately, it is not trivial to identify how the identified minerals formed, especially with Raman spectroscopy. We aim to identify the microstructural location and mineralogy of inclusions, deciphering the formation, possible transport and depositional histories require completely different methods and is far beyond the scope and possibilities of this study (and thus not mentioned as an objective). This led us to the discussion of the "state of science", it is crucial to improve the systematic combination of different methods and analysed scales. The results of this study indicate the need to investigate this complex subject further, i.e. by performing dedicated studies on chemical reactions inside the ice.
As described below in more detail we enhanced the manuscript regarding the formation of minerals and added a new section on it. To clarify this, we now finish the introduction as follows:
**(l. 103)** "Possible formation pathways of the observed minerals are briefly discussed, but are not the main focus of this study."

**Ref1:** The text also needs to say more about the rare minerals. How confident are the authors that they really found jacobsite, pyromorphite, kröhnkite, etc.? These are not common minerals, so this could be misidentification of the spectral patterns. If the authors are confident, then some sort of discussion is needed about whether these are detrital (i.e. you got lucky and found a rare mineral in dust) or authigenic, representing some sort of chemical pathway found specially in the ice core.
**Reply:** The mentioned minerals are indeed not as common as quartz or feldspar. They nevertheless have been found on all continents (see https://www.mindat.org). These minerals are assumed to be transported with dust and are thus detrital. This is one of the vastest studies about the mineralogy of micro-inclusions in polar ice0. We are thus confident that it is possible to find them in the EGRIP ice core, especially at the investigated depths with relatively high particle numbers.
From a mineral perspective, the occurrence of these minerals is sometimes supported by the associated mineralogy (RRUFF database, handbook of mineralogy). For example, jacobsite is often associated with hematite, which is mirrored in sample 2 (1 jacobsite, 11 hematite). Still, the very low absolute and relative numbers of pyromorphite, epidote, jacobsite and prehnite (they only account for 7 of all 791 identified inclusions) do not interfere with our overall study result on the mineralogy.

In summary, we conclude that these rare minerals are transported onto the ice together with more common dust minerals and add the following text to the discussion **(l. 328)**:
"The secondary minerals pyromorphite and epidote were each only found once. It is likely that they are of detrital origin and were transported together with more common dust minerals. The micrometre size of the inclusions indicates a distant source region comparable to the source regions of dust, such as the deserts of Asia (Bory et al., 2003; Vallelonga and Svensson,2014). Pyromorphite is usually found in the oxidised zones of lead ore deposits. Epidote is abundant in e.g., schistose rocks or marble. It can be a product of the hydrothermal alteration of minerals in igneous rocks, such as feldspar or mica. Prehnite was identified three times and is normally found in Mg- or Fe-rich (i.e. mafic) volcanic rocks and can be a product of hydrothermal alteration in chasms or veins. Jacobsite was found twice (S2 and S5) and can occur as a primary phase or as a product of the metamorphism of manganese deposits and thus, altered manganese minerals. It is typically associated with, among others, hematite, both were identified in S2.".

**Ref1:** In short this is a study in search of a research question. If the authors think they have answered a scientific question, they need to state the question and state the answer. If they don't think they've learned anything from their work, they should fold this data into part one.
**Reply:** As from above comments we understand that the major focus of this study was not made clear. We changed the introduction (see above) and the discussion (see below). Research questions are now defined after **l. 86**.

**Ref1:** Line 3: "Poorly understood" is a catch-all phrase. Can you be more specific about the gap in knowledge you are trying to address?
**Reply**: We added the following text to be more specific **(l. 1)**:
"Impurities in polar ice do not only allow the reconstruction of past atmospheric aerosol concentration, but also influence the physical properties of the ice. However, the localisation of impurities inside the microstructure is still under debate and little is known about the mineralogy of solid inclusions. In particular, the general mineralogical diversity throughout one ice core and the specific distribution inside the microstructure is poorly investigated, the impact of the mineralogy on the localisation of inclusions and other processes is thus hardly known."

**Ref1:** Line 3: Should "Continuous Flow Analysis" be capitalised?
**Reply:** There are publications using the capitalized version, such as McConell et al. (2006), Eichler et al. (2019), and it is thus a question of style. We can change it, but leave the final decision to the editor.

**Ref1:** Line 19: This sentence needs to be revised – the double usage of "chemical compounds" is awkward and confusing.
**Reply:** We deleted the double usage of chemical compounds **(l. 24)**.

**Ref1:** Line 20: "Also"? Is this different from their "atmospheric, marine, terrestrial or biological origin"?
**Reply:** We deleted "also" **(l. 24)**.

**Ref1:** Line 22: Peroxide needs a citation.
**Reply:** We added Legrand and Mayweski (1997) **(l. 27)**.

**Ref1:** Line 22: This sentence is fairly tautological.
**Reply:** We changed the sentence to **(l. 27)** "Insoluble impurities are rejected from the ice lattice (Ashby, 1969; Alley et al., 1986) and range in size from sub-micrometre to hundreds of micrometres (Steffensen, 1997; Wegner et al., 2015; Simonsen et al., 2019)."

**Ref1:** Line 26: Ferrosilite, wollastonite, and troilite are not dected components of the dust!
**Reply:** We changed the examples to detected components of dust **(l. 31)**:"…chemical compounds, such as $SiO_2$ or $Fe_2O_3$".

**Ref1:** Line 27: Are salts a subject of this study or not? In the previous sentence, you said it was mainly mineral dust.
**Reply:** Salts, such as sulphates or nitrates, are a subject of this study. In the previous sentence we said "mainly consist of mineral dust", salts are thus not excluded in the addressed sentence. However, we changed the

sentence to **(l. 30)**: "Water-insoluble impurities normally originate from terrestrial sources and mainly consist of salts (e.g., sulphates and nitrates) and mineral dust which is abundant in elements from the crust, such as…" To be clearer we changed "insoluble" to "water-insoluble" throughout the text.

**Ref1:** Line 29: The text about the ice lattice needs to be moved upward to where you previously discuss the ice lattice.
**Reply:** We moved it upwards as an introductory sentence to impurities, it now starts at **l. 27**.

**Ref1:** Line 31: Separate the tephra and the sulfate into different sentences.
**Reply:** We changed it to **(l. 36)**:" Tephra layers provide a record of local and distal volcanic eruptions and offer a chronological correlation between ice cores. The sulphate record can provide time lines and information about the atmospheric impact of volcanic eruptions. Furthermore, the isotopic and chemical composition of impurities enables the investigation of climatic and atmospheric processes of the past."

**Ref1:** Line 34: You already said this.
**Reply:** We changed the sentence to **(l. 40)**:" Impurities in polar ice can be measured with a variety of methods, each with certain limitations."

**Ref1:** Line 37: What is CFA? (acronym used without definition)
**Reply:** We fixed this issued and define CFA now already in **l. 43**.

**Ref1:** Line 43: What do you mean by mechanical properties?
**Reply:** By mechanical properties we refer to studies by e.g., Jones & Glen (1969) investigating creep behaviour of pure and "doped" ice at different temperatures, activation energy of creep, dislocation velocity, stress-strain curves etc.
To clarify this we changed the text to **(l. 63)**: "and mechanical properties (e.g., creep behaviour, dislocation velocity)".

**Ref1:** Line 44: Is this another area or a consequence of the control of impurities on mechanical properties?
**Reply:** This is of course related, we thus changed the text to **(l. 65)**:
"Several studies investigated the impact of impurities on the deformation of ice (e.g., Jones and Glen, 1969; Petit et al.,1987; Iliescu and Baker, 2008; Eichler et al., 2019), which in turn influences the flow of ice - a major uncertainty regarding future projections of ice sheet behaviour and solid ice discharge."

**Ref1:** Line 53: Say something more. Important for what?
**Reply:** We changed this sentence to **(l. 75)**:
"The mineralogy of impurities, their localisation and the possible interplay between both certainly plays a role in the interaction of impurities and ice physical properties (e.g., Jones and Glen, 1969; Paterson, 1991) and ice dynamics. Reasons for the current lack in understanding of the whereabouts of impurities in polar ice are manifold, the methodology to systematically look in parallel at localisation (as a microstructural feature) and mineralogy (as a chemical feature) of micro-inclusions has just recently been developed (e.g., Eichler et al., 2019)."

**Ref1:** Lines 68-72: Maybe rewrite this to be less negative about previous work.
**Reply:** We changed this to **(l. 58)** "These studies are demanding in time and costly and thus often rely on a small number of samples taken at arbitrary depths. The number of identified inclusions is thus limited and the underlying processes are not fully understood yet. Developing generalisations of the mineralogy and localisation of impurities in polar ice is hence still challenging (Stoll et al., 2021).

**Ref1:** Line 80: What companion study? Where is the citation?
**Reply:** With both preprints being available we are now able to add the proper citations in both manuscripts, which was not possible at the time of submission.
"Companion paper" has been changed to Stoll et al. 2021a throughout the text.

**Ref1:** Line 91: Unique is a dangerous word to use.
**Reply:** Indeed it is, however EGRIP is the very first deep ice core from an ice stream. We changed the wording to **(l. 109)** "an exclusive possibility".

**Ref1:** Lines 144-152: This should probably have some citations.
**Reply:** We added the following citations (**l. 165**): "The Raman effect is the inelastic scattering of light caused by the excitation of vibrational modes of crystals or molecules. This results in the Raman shift – a loss of scattered light energy, which is specific for each vibrational mode (Raman and Krishnan, 1928). The Raman shift is used to identify chemical impurities in samples in a non-destructive way – Raman spectroscopy. It is well suited for light-transparent materials, such as ice (e.g., Fukazawa et al., 1998; Weikusat et al., 2012; Eichler et al.,2019)."

**Ref1:** Lines 165-176: I would move this paragraph upwards. Some of the prior details about which peaks were distinguishable etc. could be moved to results.
**Reply:** We agree and moved this paragraph upwards to **l. 174**. We prefer to keep the details about peaks etc. in the methods and leave this choice to the editor.

**Ref1:** Line 195: Can these impurity maps be made available in a supplemental file?
**Reply:** Impurity maps are available as original .tiff files on PANGAEA. They are also discussed in more detail in Stoll et al. (2021a).

**Ref1:** Line 203: Why not include all 26 species? It's not like 26 lines is too long for a table.
**Reply:** In this table we want to focus on the most abundant species allowing some statistics, e. g., regarding grain boundaries. A table of all 26 species is indeed interesting, but was assumed to be repetitive to Figure 4 and the text. We keep the table on the most abundant minerals and provide a table with all species in the supplement (see below) (**p. 25**).

**Table A1.** Raman spectra in EGRIP Holocene ice.

| Mineral | Number | Formula |
|---|---|---|
| Gypsum | 170 | $CaSO_4 * 2H_2O$ |
| Quartz | 126 | $SiO_2$ |
| Mica undefined | 42 | $(K, Na, Ca, NH_4)Al_2(Si_3Al)O_{10}(OH)_2$ |
| Feldspar | 67 | $(K, Na, Ca, NH_4)(Al/Si)_4O_8$ |
| Mg-sulphate | 64 | $MgSO_4$ |
| Bloedite | 61 | $Na_2Mg(SO_4)_2 * 4H_2O$ |
| Nitrates | 36 | $XNO_3$ |
| Hematite | 33 | $Fe_2O_3$ |
| K-sulphate | 31 | $K_2SO_4$ |
| Krohnkite | 27 | $Na_2Cu(SO_4)_2 * 2H_2O$ |
| Mica (Phlogopite) | 25 | $KMg_3(Si_3Al)O_{10}(OH, F)_2$ |
| Sulphate undefined | 24 | $XSO_4$ |
| Carbonous | 22 | C |
| Dolomite | 17 | $CaMg(CO_3)_2$ |
| Mica (Muscovite) | 25 | $KAl_2(Si_3Al)O_{10}(OH, F)_2$ |
| Na-sulphate | 9 | $NaSO_4$ |
| Anatase | 8 | $TiO_2$ |
| Rutile | 3 | $TiO_2$ |
| Mica (Paragonite) | 3 | $NaAl_2(Si_3Al)O_{10}(OH)_2$ |
| Prehnite | 3 | $Ca_2Al_2Si_3O_{10}(OH)_2$ |
| Na-nitrates | 3 | $NaNO_3$ |
| Jacobsite | 2 | $MnFe_2O_4$ |
| Titanite | 2 | $CaTiSiO_5$ |
| Epidote | 1 | $Ca_2(Fe/Al)Al_2(Si_2O_7)(SiO_4)O(OH)$ |
| Pyromorphite | 1 | $Pb_5(PO_4)_3Cl$ |
| Air | 1 | $O_2$ |
| Undefined | 17 | ? |

**Ref1:** Figure 4: The color scheme for this figure is strange. I would cluster the minerals by type rather than alphabetically. The color coding needs to relate to how you discuss minerals later in the text. I.e. all minerals that you presume to source from dust need to be clumped together and similarly colored. Sulfates and nitrates and black carbon should each be their own unique colors that cannot be confused with other minerals. Similar minerals should be next to each other with similar colors. I.e. rutile and anatase, "mica" and prehnite, etc.
Also, why is graph a normalised to 100% but graph b presents absolute values.
Also, might it be possible to have time on the top axis to counter depth on the bottom axis? And, I think two decimal points for depth is excessive.
**Reply:** We agree that Fig. 4 can be displayed in a clearer way and appreciate the specific comments. We tried out different colour codes (while keeping it colourblind-friendly) which turned out to be quite difficult due to

the amount of different mineral types.

We applied your input and changed the figure **(p. 12)**. Minerals are now grouped regarding their mineralogy/source, there is however a limitation to the possibilities in representing all mineral species while keeping a colourblind-friendly and scientifically useful (Kramer et al., 2020) colour code.

A) and B) both display information about the absolute values (n=), but there is a difference. A) shows all identified inclusions while b) only shows the identified sulphates.

Showing the absolute values in B) helps to visualize and clarify that there are other, non-sulphate minerals at these depths too. Showing two plots, each displaying 100%, could be confusing. However, we can change it to relative amounts if the editor decides this might be better.

We added a new Fig. 1, which shows time on the y-axis. The samples are now called S1-S11 increasing readability (see answer to referee 2).

[Figure]

Figure 1 Identified Raman spectra of micro-inclusions in EGRIP Holocene ice, n is the total amount of identified spectra per sample. The two deepest samples are from the Younger Dryas and Bølling Allerød and have by far the highest water-insoluble particle concentrations. A) All identified Raman spectra per sample. For better visibility some Raman spectra are condensed in groups (e.g., sulphates and mica). B) Identified sulphates in detail. Sulphate diversity decreases below 900 m.

**Ref1:** Table 2: Please clarify why some species are identified by mineral name and others aren't. Is it unclear which magnesium sulfate mineral is present, for instance?

**Reply:** This is clarified in the main text, but it is a good idea to add such an explanation in the table header. The exact mineral couldn't be identified **(p. 11)**: "Mg- and K-sulphate minerals could not be identified clearly."

**Ref1:** Line 209 (and elsewhere): Terms like carbonate and sulfate can be confusing when applied in the fashion. It's probably better to say "the carbonate mineral ...". Though in this case, everyone should know that dolomite is a carbonate mineral, as you've included the chemical formula.

**Reply:** To avoid confusion we changed the text to **(l. 227)** "Sulphate minerals (from now on "sulphates") can be…" and **(l. 230)** "carbonate mineral dolomite".

**Ref1:** Line 210 (and elsewhere): The "might be" has me a little worried. Maybe in table 2 you can add a column for alternate possibilities for the species where identification is in doubt. Or maybe create a supplemental file that lays out the alternative possibilities.

**Reply:** As with many other methods there is no 100% certainty with Raman spectroscopy since spectra can be similar and the boundary conditions (especially in ice) have an impact on the clarity of the derived spectrum. We excluded all spectra of high uncertainty and "might be" should only clarify this small inevitableness. To avoid confusion, we deleted **(l. 231)** "might be".

Pyromorphite and epidote were only found once, jacobsite only twice. It is thus impossible to compare their spectra with many other examples as normally done. As discussed above, they only represent a tiny fraction of the overall mineralogy and discussing them in detail would evoke a wrong impression. Table 2 is already very wide, we thus leave it like it is.

**Ref1:** Line 267: In what sense is this data accompanying?

**Reply:** We changed the sentence to **(l. 286)**: "Applying different methods (CFA, microstructure-mapping and Raman spectroscopy) enabled us to create high-resolution impurity maps showing that sulphates and terrestrial dust are the most abundant minerals."

**Ref1:** Line 321: This is confusing to me. How is going from sulfates and dust to dust and sulfates (i.e. gypsum) a meaningful shift?
**Reply:** We find a change in mineralogy with depth. First, the occurrence of sulphates is changing: with different types of sulphates in the upper part and only gypsum in the lower. Second, we do find dolomite and nitrates only in the lower part, but not in the upper. These findings indicate a different mineralogy in these two intervals. Using the wording "shift" can be misleading, we thus replaced it with **(l. 334)** "change in mineralogy": „We have observed a pronounced change in mineralogy from a sulphate-diverse, while terrestrial dust rich, regime in the upper900 m to a terrestrial dust-dominated regime with partially high amounts of gypsum (Fig. 5)."
**(l. 347)** "Even though we find this difference in mineralogy between the upper and lower part, this change does not correlate with a major change of the climate period, i.e. from Glacial to Holocene ice."

**Ref1:** Line 325: What does it mean to expose mineral diversity in detail?
**Reply:** The intention was to strengthen the previous sentence, stating the findings support and deepen the previous ones. We deleted this sentence.

**Ref1:** Line 326: You already said this in the previous paragraph.
**Reply:** We have restructured the discussion to clarify the different aspects of our findings. We hope this will clarify the content of each paragraph. The mentioned sentence was changed to **(l. 340)**: "The most common minerals, i.e. sulphates and terrestrial dust, are found throughout the entire ice core, but in varying abundance (Fig. 5 and simplified in Fig. 9)."

**Ref1:** Line 330: This makes no sense in terms of charge balance and stoichiometry. Sulfate can't just blow in without any kind of attached cation. It could be deposited as acidic aerosols of sulfuric acid (i.e. either H2SO4 or SO3) and then gain its cations from other sources. Or it is deposited as a sulfate-bearing compound (gypsum, etc.). You need to explain the full chemical sequence of events that you are proposing. If it was acidic and then exchanged with chloride salts and precipitated, you would be left with a highly acidic chloride rich brine (i.e. HCl), probably in the grain boundaries of the ice crystals. If you really think that there is HCl in ice cores, by all means defend that point.
**Reply:** We think this is a misunderstanding, we do not state that sulphates just blow in without a cation. Ohno et al. (2005, 2006) suggest that the process you describe could occur in ice cores resulting in Cl and HCl entering the ice lattice and creating point defects. Unfortunately, Raman spectroscopy is not able to measure HCl in solid ice since HCl is expected to be a solution of monoatomic ions without vibrational spectra. Cl however was found in ice in several studies (e.g., Barnes et al. 2002a, Cullen and Baker, 2002, Obbard et al., 2003). Ohno et al. (2005, 2006) and we can thus only mention that these reactions could be possible since we (and them) found several sulphates with the cations $Na^+$ or $Mg^+$. This can be explained by the difference in $Ca^{2+}$-concentration in the atmosphere during the Holocene and the Glacial as discussed by Röthlisberger et al. (2003). They state that acid-gas particles, such as $H_2SO_4$, reacted with sea salt aerosols during transport and in the snowpack while the acids were neutralized by terrestrial dust, i.e. $Ca^{2+}$, in the Glacial.

We changed the paragraph to clarify this:
**(l. 404)** „The high abundance of sulphates in many samples can be largely explained by blown-in sulphate-bearing compounds, such as gypsum. However, it is suggested that sulphate minerals can also form after initial deposition when acidic aerosols of sulfuric acid, such as H2SO4 droplets, gain cations from other sources (e.g., Ohno et al., 2005). The main cations of the sulphate minerals in our samples are Ca2+ ,Mg2+, and Na+(Table 2). Ohno et al. (2005) suggest that Mg2+, and Na+ could originate from blown-in sea salt (NaCl and MgCl2), which was deposited on the ice sheet. This would result in the possibility that HCl and Cl−enter the ice lattice creating point defects and thus effecting the electric conductivity (Petrenko and Whitworth, 1999; Ohno et al., 2005) and dislocation mobility (e.g., Hu et al., 1995). HCl solution is monoatomic and without a vibrational spectrum and thus not detectable with Raman spectroscopy. We are thus limited in evaluating the process of sulphate formation, but our results imply that the process proposed by Ohno et al. (2005) could occur. However, deeper samples show smaller amounts, and varieties, of sulphates, which almost completely disappear below 900 m. In the four deepest samples gypsum is the only sulphate, except for one Krohnkite micro-inclusion in S9. Data

was scarce until now, but e.g., Sakurai et al. (2009, 2011) and Eichler et al. (2019) show that sulphates are found in deeper parts of ice sheets, also consisting almost entirely of gypsum supporting our results. One possible explanation is that during the Holocene sea salt aerosols reacted with acid-gas particles during transport and in the snowpack. In the Glacial however acids were often neutralised by the higher amounts of terrestrial dust, i.e. with high concentrations of $Ca^{2+}$, and thus reacted less with sea salt (Röthlisberger, 2003). Fittingly, our results show that high dust concentrations correlate with the dominance of gypsum over other sulphate minerals while lower dust values correlate with medium to high Na- and Mg-sulphate."

**Ref1:** Line 334: Is data scarce? This paper certainly made it less scarce.
**Reply:** Thank you. We changed the sentence to **(l. 345)**: "Data were scarce until now, but…"

**Ref1:** Line 345: A closer investigation? Were they investigated at all?
**Reply:** We started investigating samples from the Glacial, but this is not part of this manuscript.
We thus changed the sentence to **(l. 354)**:" An investigation of EGRIP glacial samples will show if prominent changes in mineralogy also occur in deeper depth than the analysed upper 1340 m."

**Ref1:** Section 4.2.2: This might benefit by some sort of figure where you plot your data against that of Sakurai et al.
**Reply:** This is very true and we would like to follow up on this. However, the data available online only shows the total amount of inclusions analysed with Raman spectroscopy (which sometimes differs strongly from the number of identified inclusions). Plotting our data against their data is thus unfortunately not possible.

**Ref1:** Line 349: Awkward writing.
**Reply:** We agree and changed the sentence to **(l. 358)**: "In Greenland, the GRIP ice core was analysed with Raman spectroscopy and EDS by Sakurai et al. (2009). The GRIP and EGRIP drill sites are located relatively close by, we thus compare both cores here in detail."

**Ref1:** 359: Didn't you intentionally select dust rich sections? Did Sakurai et al. do the same? If not, there's your explanation.
**Reply:** Unfortunately, this is not mentioned in Sakurai et al. (2009). They only mention the depth and age of their samples. We are thus not able to check this hypothesis, but this could be a part of the explanation.
We added it in the following way **(l. 369)**: "The higher amount of terrestrial minerals in our samples is difficult to explain conclusively, because the GRIP and EGRIP sites are comparably close to each other and in similar distances to the coast. Intentionally choosing samples with a high insoluble particle content could partly explain this observed difference.".

**Ref1:** 374: I think you actually have a good number of observations. I don't think this section needs to be as self-critical.
**Reply:** Thank you, we changed this section to **(l. 384)**: "Some Raman in S3 have been difficult to identify, we thus had to exclude several spectra."

**Ref1:** Line 380: What about XRD? That's usually quite a reliable way to do mineralogy.
**Reply:** This is true, but XRD does not provide information about the location of the inclusions, which SEM and LA-ICP-MS do. We thus suggested these two methods.
We clarified this by changing the sentence **(l. 390)**: "A partial solution would be an analysis of the same samples with a subsidiary method which identifies location and chemistry of impurities, such as SEM coupled with EDS or LA-ICP-MS."

**Ref1:** Line 385: I don't understand the methodological explanation.
**Reply:** We deleted this part of the sentence since we discuss the issue of statistics in a different paragraph. The text is now **(l. 465)**: "This can partly be explained by different deposition conditions or by chemical reactions taking place in the ice.".

**Ref1:** Line 388: Iizuka et al. say that the sulfatisation is proportional the dust flux, which makes sense because it gives the atmospheric sulfuric acid something to react with.
**Reply:** This is true, to clarify we changed the text to **(l. 415)**:

"Iizuka et al. (2012) concluded that aerosol sulfatisation in Antarctica is proportional to the dust flux and more likely to occur in the atmosphere or during fallout than after deposition."

**Ref1:** Line 394: Why would dry deposited sulfates have such diverse mineralogy? Where are the sulfates coming from? The question isn't really explained or addressed.
**Reply:** We here refer to the finding of high sulphate concentration in relation to wind crusts, indicating dry deposition of sulphate (Hoshina et al., 2014, Moser et al., 2020). Assuming that in periods of no accumulation (i.e. period of crust formation) sulphate aerosols are accumulated from dry deposition at the surface in layers with high concentration and get buried with time. We speculate, that these layers formed at the surface can be one source of the observed concentration of sulphates in deep ice. In the addressed sentence dry deposition is not suggested as a direct cause for mineral diversity, but as an explanation for the clustering and layering of sulphates in some samples ("The strong layering and clustering of sulphates observed in some samples could originate from dry deposition events, which form deposition crusts mainly containing sulphates.")

**Ref1:** Line 398: There's quite a lot of reference to future work here. I would rather have you focus on what you can infer from this work.
**Reply:** We restructured the discussion to strengthen our findings as shown with examples within this reply. We merge and sharpen section 4.3 and 4.4 into one section, which enables a stronger focus on our work. With the increasing number of publications related to EGRIP, referring to findings get easier. We now e.g., can include the study by Gerber et al. (2021) in the methods and discussion. Showing the new structure is not possible within this reply, but below are some examples:

**(l. 485)** "A difference in the chemistry at the ice sheet surface at the time of deposition and different atmospheric circulation patterns, and thus varying aerosol input, could be indicated by the micro particle record (Fig. 1). Dust particle numbers generally increase with depth (Fig. 2) and especially S10 is from a dust-rich period, the Younger Dryas."

**(l. 116)** "Accumulation rates were highest 7.8 kyr ago (0.249 ma−1) and decreased towards the Last Glacial Period with a peak during the Bølling Allerød. Due to the flow of NEGIS ice from the last 8 kyr was deposited under increasingly higher accumulation rates with increasing age caused by higher precipitation closer to the ice divide (Gerber et al., 2021)."
**(l. 124)** "Gerber et al. (2021) propose that Last Glacial Period ice was deposited 197 to 332 km upstream from EastGRIP."

**(l. 494)** "Our analysed samples were deposited within 197 km upstream from EGRIP and thus at slightly higher surface elevations (2993±7 m a.s.l. at 1400 m depth) (Gerber et al., 2021), which limits the impact on the aerosol input. Accumulation rates for ice from depths of 900 to 1400 m were low, except the peak during the Bølling Allerød. This peak coincidences with the high $I_{var}$ value in S11 displaying that a high accumulation rate enhances mineral diversity in this climatic period, contrary to the Holocene. However, it is difficult to compare the Holocene samples to the two samples from the Younger Dryas and Bølling Allerød. A systematic follow-up study on EGRIP Glacial ice is needed to investigate if the observed trends, e.g., of mineral diversity, continue with depth."

**Ref1:** Line 404: What do Fe minerals have to do with anything? I don't follow the logic of this paragraph. I.e. are you claiming that Fe is a sign of authigenic mineral formation? If so, which minerals, which reaction pathways?
**Reply:** We here aim at discuss different studies investigating this idea and Fe-bearing minerals and their reactions inside the ice have been analysed in several publications recently. The recent works by Baccolo et al. investigate possibilities of ice as a geochemical reactor using Fe-minerals as examples (see Baccolo et al. 2018 and Baccolo et al., TCD). Baccolo et al. (2021) discuss the formation of jarosite in the Talos Dome ice core and interpret it as a product of weathering involving acidic atmospheric aerosols and dust.
Since we found Fe-minerals at several depths this is of direct relevance. Contrary to Baccolo et al. (2018, TCD) our method and data do not aim to investigate reaction pathway. An in-depth discussion of the reaction pathways or mineral formation is thus not possible, but it would be wrong to intercept the on-going progress in this field. Similar in-depth studies would be needed to investigate the possible reactions occurring in ice, this goes well beyond the scope of this study.

To clarify this we changed the text to **(l. 439)**: "Fe-minerals, such as hematite and jacobsite, were identified in this study and occur relatively often in polar ice (e.g., Baccolo et al., 2018; Eichler et al., 2019; Baccolo et al.,

2021). Baccolo et al. (2018) showed that the deep ice sheet environment is not anoxic and dissolved oxygen and liquid water veins might support the oxidation and dissolution of specific mineral phases. Investigating such specific chemical reactions of Fe-minerals is beyond the scope of this study, but they display the lack of knowledge regarding ice as a chemical reactor. Furthermore, Faria et al. (2010) observed the formation of solid inclusions in deep ice, which was supported by Eichler et al. (2019). A local mixing of impurities in shear bands with high strain rate and strain could explain our observations of clustering of sulphates and is not unlikely due to the dynamic conditions inside NEGIS. Local small-scale processes could lead to preferred clustering of micro-inclusions with similar chemistry. We observed preferred clustering at all depths, however samples below 900 m show significantly fewer clusters. This correlates with the depth of declining sulphate-diversity and could indicate that, around this depth, certain unknown chemical reactions occur or that large-scale boundary conditions, such as climate or ice sheet extent, changed during the time of deposition".

**Ref1:** Line 418: Typography
**Reply:** Corrected.

**Ref1:** Line 431: What is the point of this whole paragraph if you only conclude that this unlikely to be relevant and untested?
**Reply:** In order to include the possible upstream effect on our findings we include the topic. For clarity, the section was re-arranged and work by Gerber et al. (2021) was included (see above).

**Ref1:** Line 435: What do you mean it is unlikely? Either it is observed, or it isn't. This constant reference to non-existent studies is really grating.
**Reply:** We refer to studies by eg., Svensson et al. (2000), Bory et al. (2003), Mojtabavi et al. (2020) and do a detailed comparison with the results from Sakurai et al. (2009) in Sect. 4.2.2. The recent publication of Gerber et al. (2021) addressing the upstream effect of EGRIP has been included in the text as well.

**Ref1:** Section 4.5: This entire section is pointless. There are no research questions addressed here.
**Reply:** We deleted this section.

**References:**
McConnell, J. R., Lamorey, G. W., Lambert, S. W., and Taylor, K. C.: Continuous Ice-Core Chemical Analyses Using Inductively Coupled Plasma Mass Spectrometry, *Environmental Science & Technology*, 36, 7–11, https://doi.org/10.1021/es011088z, 2002.
Eichler, J., Weikusat, C., Wegner, A., Twarloh, B., Behrens, M., Fischer, H., Hörhold, M., Jansen, D., Kipfstuhl, S., Ruth, U., Wilhelms, F., and Weikusat, I.: Impurity Analysis and Microstructure Along the Climatic Transition From MIS 6 Into 5e in the EDML Ice Core Using Cryo-Raman Microscopy, *Front. Earth Sci.*, 7, 1–16, https://doi.org/10.3389/feart.2019.00020, 2019.
Moser DE, Hörhold M, Kipfstuhl S and Freitag J (2020) Microstructure of Snow and Its Link to Trace Elements and Isotopic Composition at Kohnen Station, Dronning Maud Land, Antarctica. *Front. Earth Sci.* 8:23. doi: 10.3389/feart.2020.00023
Hoshina, Y., Fujita, K., Nakazawa, F., Iizuka, Y., Miyake, T., Hirabayashi, M., et al. (2014). Effect of accumulation rate on water stable isotopes of near-surface snow in inland Antarctica. *J. Geophys. Res*. 119, 274–283. doi: 10.1002/2013JD020771
Gerber, T. A., Hvidberg, C. S., Rasmussen, S. O., Franke, S., Sinnl, G., Grinsted, A., Jansen, D., and Dahl-Jensen, D.: Upstream flow effectsr evealed in the EastGRIP ice core using Monte Carlo inversion of a two-dimensional ice-flow model, *The Cryosphere,* 15, 3655–3679, https://doi.org/10.5194/tc-15-3655-2021, 2021.
Baccolo, G., Cibin, G., Delmonte, B., Hampai, D., Marcelli, A., Stefano, E. D., Macis, S., and Maggi, V.: The Contribution of Synchrotron Light for the Characterization of Atmospheric Mineral Dust in Deep Ice Cores : Preliminary Results from the Talos Dome Ice Core (EastAntarctica), *Condensed Matter*, 3, https://doi.org/10.3390/condmat3030025, 2018.
Baccolo, G., Delmonte, B., Niles, P. B., Cibin, G., Di Stefano, E., Hampai, D., Keller, L., Maggi, V., Marcelli, A., Michalski, J., Snead,C., and Frezzotti, M.: Jarosite formation in deep Antarctic ice provides a window into acidic, water-limited weathering on Mars, *Nature Communications,* 12, 1–8, https://doi.org/10.1038/s41467-020-20705-z, http://dx.doi.org/10.1038/s41467-020-20705-z, 2
Stoll, N., Eichler, J., Hörhold, M., Erhardt, T., Jensen, C., and Weikusat, I.: Microstructure , Micro-inclusions and Mineralogyalong the EGRIP ice core - Part 1: Localisation of inclusions and deformation patterns, *The Cryosphere Discussions*, pp. 1–29,755https://doi.org/https://doi.org/10.5194/tc-2021-188, 2021
Jones, S. J., and Glen, J. W. (1969). The effect of dissolved impurities on the mechanical properties of ice crystals. *Phil. Mag.* 19, 13–24. doi:10.1080/ 14786436908217758

https://www.mindat.org, last access 24.09.21 08:51
https://rruff.info/doclib/hom/jacobsite.pdf, last access 23.09.21 16:36

**Replies to referees**
**Referee 2**

**Ref2:** The manuscript presents a series of detailed investigations of cm-scaled sections from the EGRIP ice core, mostly from the Holocene ice core section. The sections are analyzed for cryo-Raman spectra, microstructure, microscopy, and compared to high-resolution dust records. Based on this, the temporal development of the ice core mineralogy is discussed.
The manuscript (MS) is very detailed almost providing a review in some section, figures are generally good, and referencing is satisfactory. The analytical work presented in the paper is impressive and of relevance to the community. I have some comments and concerns in the following.

**General comments**
**Ref2:** Whereas the MS discusses to a great detail how the impurity composition results may relate to the long-term climatic context, I think another important property/sample characteristics is somewhat overlooked, namely the sample seasonality**.** Because almost all of the impurities in the Greenland ice sheet show some kind of seasonal pattern/variation, I think the season from where your samples are taken may influence their composition just as much as the longer term climate context (eg where in the Holocene the sample is taken). For example concerning dust, we know that today we have a large Asian-derived dust spike in Greenland in spring/summer, whereas the dust in other seasons could be of different origin(s). The detailed sample composition may therefore depend strongly of which season the sample is take from. In Fig. 1 (b)-(l), we see that all of the samples appear to be associated with a dust spike. Does that mean that all samples are from spring/summer? I think that with all of the high resolution profiles that are available for EGRIP, it should be possible to determine the approximate seasonality of the samples? For the Holocene, you may for example compare the CFA profiles to those of (Gfeller et al., 2014).
Likewise, it may be of use to make more comparison to the CFA / DEP / ECM record or to the line scan profile across the sections you have sampled. Are samples with high sulfate associated with high DEP/ECM/Conductivity? Are you in a winter layer with high NaCl? Are the dust concentrations typical? Is here a possibility that one or several samples coincidence with forest fires (NH4), a volcanic eruption (DEP/ECM/Conductivity) or some other atypical feature?
One sample is from a cloudy band. Is this a 'typical' cloudy band for the period or is it somehow exceptional? It is stated that in this MS you 'focus on the chemistry', but there is very little chemistry data shown and the comparison between 'chemistry' and the Raman and other results is sparse.
**Reply:** We thank the referee for a detailed review and several good ideas, which help to put our work in context and to enhance readability. The majority of the suggestions were adopted and are discussed below. You present several good ideas, but not all of them are feasible within this study and would require dedicated personnel and specific studies (e.g., on the seasonal changes of certain parameters). The general idea of our study is to give an overview of the mineralogy of inclusions in EGRIP Holocene ice, tackling very specific features of more records is thus not expedient.

CFA and Raman spectroscopy data were both categorized as chemistry data in this context. To clarify this we changed the text to **(l. 139)** "we here focus on the mineralogy and insoluble particle content." To clarify this throughout the text we specified the terms of data we use throughout the manuscript. We included the DEP and ECM records (new Fig. 1 below) which provide a good overview since water isotope data are not available yet. We indicate the depths of the analysed samples in the dust record and also show an age scale.
We did a high-resolution analysis of conductivity and acidity at the analysed depths, but no prominent features were found (Figure below). The most prominent one is a small wavy increase of acidity from 757.18 to 757.24 m (sample 6). This is also related to the fact that sample 6 is the longest sample and a thus higher chance of a change in acidity or conductivity. The more acidic part is characterized by different sulphates as displayed in Fig. 7a. No seasonality signal or clear correlations are visible from this plot and we thus do not include them. This is potentially due to different reasons. The main reason is probably the difference in resolution of the applied methods (DEP every 5 mm, ECM every 1 mm, inclusions for Raman a few micrometer) again demonstrating the challenge in combining different scales and methodological approaches (continuous vs. discrete). Another aspect is the slight difference in the measured planes of the samples. The same applies to Visual Stratigraphy data, deciphering clear signals is difficult on such a small scale.
We did not find any indications for an "exceptional cloudy band", but you raise an interesting point. There is work in progress regarding the cloudy bands in EGRIP ice and Raman spectroscopy might play a role in this.

[Figure]

Figure 2 Number counts of particles larger than 1μm per ml of melt water (dust) derived via CFA from the upper 1350 m of the EGRIP ice core. Samples analysed with Raman spectroscopy are indicated with arrows. Acidity data from Mojtabavi et al. (2020b), conductivity data from Mojtabavi et al. (2020c). Age from Mojtabavi et al. (2020a), b2k = before 2000 C.

[Figure]

**Ref2:** I think the authors need to spend a little more time working with the wording of the text. In every other line, I think there are imprecise statements or the wording is not concise. I gave up making a list of specific places where I think the text could be improved, as I think this is a task of the authors.

**Reply:** We enhanced the wording throughout the manuscript, it is now more concise and precise. Examples are found throughout the text, this reply, and in the reply to referee 1.

**Specific comments**

**Ref2:** Throughout the MS there is reference to 'bags'. Whereas this may be a meaningful notation for those working on ice cores on an everyday basis it may not be the most obvious notation for the reader. I would suggest to replace with '55 cm sample' or similar throughout the MS.

**Reply:** We agree and changed "bag" to **(e.g., l. 134, 292)** "55 cm sample" throughout the text.

**Ref2:** Throughout the MS there is reference to 'a companion paper'. Rather than making the reader start guessing about what paper that might be, I suggestion to cite to the full reference.
**Reply:** As stated for Referee 1 at the time of the submission there was no proper way to cite the companion paper, because of the submission at the same time. Since part 1 and part 2 now have a citable doi etc. we changed "companion paper" to "Stoll et al. (2021a)".

**Ref2:** In several places there is mentioning of the upstream effects at EGRIP. There is now a paper discussing those effects at EGRIP and it may be discussed how important upstream effects may be for the sampled intervals (Gerber et al., 2021).
**Reply:** We are happy that this paper is published and refer to it in the methods and the discussion:
**(l. 358)** "Accumulation rates were highest 7.8 kyr ago (0.249 ma−1) and decreased towards the Last Glacial Period with a peak during the Bølling Allerød. Due to the flow of NEGIS ice from the last 8 kyr was deposited under increasingly higher accumulation rates with increasing age caused by higher precipitation closer to the ice divide (Gerber et al., 2021)."
**(l. 124)** "Gerber et al. (2021) propose that Last Glacial Period ice was deposited 197 to 332 km upstream from EastGRIP."
**(l. 494)** "Our analysed samples were deposited within 197 km upstream from EGRIP and thus at slightly higher surface elevations (2993±7 m a.s.l. at 1400 m depth) (Gerber et al., 2021), which limits the impact on the aerosol input. Accumulation rates for ice from depths of 900 to 1400 m were low, except the peak during the Bølling Allerød. This peak coincidences with the high $I_{var}$ value in S11 displaying that a high accumulation rate enhances mineral diversity in this climatic period, contrary to the Holocene. However, it is difficult to compare the Holocene samples to the two samples from the Younger Dryas and Bølling Allerød. A systematic follow-up study on EGRIP Glacial ice is needed to investigate if the observed trends, e.g., of mineral diversity, continue with depth."

**Ref2:** For the discussion of the extend of the Greenland ice sheet in earlier periods and possible costal dust sources (l. 422-432), you may refer to (Simonsen et al., 2019).
**Reply:** We added Simonsen et al. (2019) and extended the discussion to **(l. 476)**: "The RECAP dust record shows the exposition of local dust sources, e.g., in King Christian X land, between 12.1 ± 0.1 to 9.0 ± 0.1 ka b2k (Simonsen et al., 2019).".

**Ref2: Figure 1:** This is the figure where you put your samples into a climatic context and I have a number of comment/suggestions:
- It is very important that overall climatic context is clear, therefore, it would be very helpful to include to water isotopic profile (d18O) in the figure, eg we need to know exactly where the YD onset and terminations are in relation to the samples and which part of BA your sample are taken from. If the EGRIP isotopes are not released you can show DEP or ECM or transfer the isotopes from another deep ice core.
**Reply:** This is a valuable comment to increase the accessibility for a broader community. Unfortunately, the EGRIP d18O record is not released yet, we thus plot the ECM and DEP profiles next to the dust record as explained above (see Fig. 1 above).

**Ref2:** - In (a) I do not understand what the black dots represents. How are the depths chosen? Isn't there a continuous dust profile for the Holocene? It seems like the dot density is very irregular? Considering the abrupt change in dust concentration at the YD boundaries by an order of magnitude or more, the smoothing of the dust profile appears somewhat unjustified.
**Reply:** The black dots were chosen to represent the 55 cm bag values which were analysed for physical properties, and thus available for Raman measurements. We now show a continuous dust profile of all 55 cm sample means together with continuous DEP conductivity and ECM acidity data.

**Ref2:** - In all figures it says the x-axis shows dust particles per ml. Are those the >1 micron particles only or what size fractions are included. This should be specified.
**Reply:** This is addressed in section 2.2 Continuous Flow Analysis: "Micro-particle concentrations where determined using an Abakus (Fa Klotz) Laser Particle Sizer (e.g., Ruth et al., 2003) operating in the size range between 1-15 µm, which covers the size range of optical microscopy."
To clarify this in the figure we changed the figure caption to: "Number counts of particles larger than 1 µm per ml of melt water (dust) derived via CFA from the upper…".

**Ref2:** - In all of the Greenland dust profiles I know of (Eg (Ruth et al., 2002; Schüpbach et al., 2018)), the Younger Dryas is characterized by a much high (order of magnitude) dust concentration than the Holocene, whereas the BA period has intermediate dust levels. This pattern is not at all reflected in the dust curve shown in (a). The indication of the YD interval appears inconsistent with the depths provided lines 100-103.
**Reply:** We mixed something up, the YD is deeper in the core than shown in the plot – thanks for noticing! We corrected this and the dust profile now aligns with the mentioned literature and is clearly visible in Fig. 1.

**Ref2:** - Figures (b)-(l) nicely show the position of the sample in context of the continuous dust profile, but they do not give an impression of the absolute dust level in each sample, which vary by orders of magnitude and may have important implications for the interpretation. I would suggest to either use a common log scale for all the figures or to keep-as-is but then add another column of figures that shows the absolute level at the same detailed depth resolution. The dust level at the sample resolution is a basic parameter that that may fundamentally impact the sample composition, and it is not deducible from (a).
**Reply:** We appreciate this detailed feedback and decided to split up both figures. A new figure now shows the insoluble particle number with depth (log-scale) together with acidity and conductivity data as described above (now Fig. 1). The detailed sections are now shown in a second figure (Fig. 2 below). We changed the labeling of (now) Fig. 2 to "…Please note the different scales in dust concentration on the abscissa. S10 is within a cloudy band, a horizontal layer of much higher dust content than the area above and below."

[Figure]

Figure 3 Number counts of insoluble particles larger than 1 μm per ml (dust) derived via CFA from the chosen depths analysed with Raman spectroscopy. Please note the different scales in dust concentration on the abscissa. S10 is within a cloudy band, a horizontal layer of much higher dust content than the area above and below.

**Ref2:** - Caption: please be somewhat more precise in this and other captions: Eg 'Dust data' potentially means 'Number counts of particles larger than 1 micron per ml of melt water'? '55 cm bags' may not make sense to the reader. 'Cloudy band' may not make sense to the reader. Refer to main text if explained elsewhere.
**Reply:** We edited all captions and applied the suggested changes.

**Ref2: Table 1**: Please specify what the depth refers to: top, middle or bottom of sample. If someone wants to compare your results to other records it is important to know the exact sample position. You may consider naming your samples, eg S01, S02, ... S11 rather than referring to the sample depth in the text. This may improve the readability of Figure 4 and others. If you do that, this table should include the sample names. You

may also include information about what criteria the individual samples are selected from. Furthermore, information about the mean crystal size, fraction of sample covered by crystal boundaries, the mean dust and salt concentrations and the sample average conductivity/DEP/ECM level(s). If possible, information about the season from where the sample is taken could be included as well.

**Reply:** We changed the table in several ways. We happily adopted the suggestion to use S1-S11 and changed it in the text and Table 1. Furthermore, we added the exact depths intervals of the Raman samples. The exact depths were shown in Fig. 1, adding them to Table 1 should make it easier to compare them to other records. For these exact depths we added the mean acidity and conductivity values to Table 1. Conductivity and acidity are now also displayed in Fig. 1 (see longer explanation above).

Including much more information in one table is not possible for various reasons. For example, information about the crystal size would not make sense due to poor statistics at the usual Raman sample size of ~1 x 1 cm. This could only be shown with a good statistic for the entire 9 cm sample (thin section) which would give a wrong impression of the specific area analysed with Raman spectroscopy. The "bulk" information is however included in part 1 and thus easily available (Stoll et al., 2021a).

We appreciate the other ideas, but it is not possible to do all this within this study. For example, investigating the seasonality along the core is a major task as explained above. More details regarding the DEP/ECM record are mentioned above.

**Ref2: Table 2:** You investigate the fraction of impurities that are found in grain boundaries. I think it is relevant in this context also to state the fraction of each sample that is covered by grain boundaries according to your 300 micron definition? For some samples it appears that a quite a large fraction of the analyzed area is covered by boundaries. If you then subtract the area covered by air bubbles, it could be that in the end there is no preference of the impurities to be located in a boundary or not?

**Reply:** We did not include the area occupied by the grain boundary as this is shown (and thus referred to) in Stoll et al. (2021a) and the table is already quite broad. It would be necessary to include each sample, the area occupied by inclusions and by grain boundaries, ergo a new table (which would be similar to Table 1 in Stoll et al., 2021a). We thus added a link to Stoll et al. (2021a): "Details on grain boundaries are shown in Table 1 in Stoll et al. (2021a). In the updated version of Stoll et al. (2021) we also discuss the impact of different grain boundary thicknesses (100, 200, and 300 micron).

Investigating the area covered by air bubbles is an interesting approach, but is not suitable for this study. The volume of bubbles is magnitudes larger than grain boundaries and the insecurity with depth (sample surface and inclusions 500 um below) would be large. Thus, a different methodological approach might be needed. It is definitely an interesting approach, which could be investigated in a different study.

**Ref2: Figure 4:** Based on the sulfate diversity presented in Fig. 4(b) you conclude that there is a general decrease in the sulfate diversity with depth and in the abstract you mention that there is a change at around 900 m depth that is also discussed at length in the discussion. I think this conclusion is poorly supported by the data. Indeed, the sulfate diversity of the four deepest samples is low, but it is also low for two other samples from above 900 me depth. The two deepest samples are from the last glacial period where many climatic conditions were quite different from the Holocene conditions, so I am not sure those two deeper samples are directly comparable to the younger ones. An alternative interpretation of the figure would be to say that the sample from 1062.65 m depth looks unusual in terms of sulfates, but that all of the other samples are similar, leaving out the two deepest samples that are from a different climatic period. In other words, I think the statistics may not allow for the conclusion you make.

**Reply:** We agree that it is difficult to draw definite conclusion from our statistics even though numbers and spatial-resolution are comparably large. We would like to address the two mentioned aspects separately:

1) Decrease in sulphate diversity

We mainly focus on the (almost) non-abundance of other sulphates than gypsum below 900 m. Sulphates are not consistently diverse throughout the upper 900 m, but there is usually more than one sulphate at each depth (6/7 samples have at least 2 sulphates in the upper 900 m).

You are right, there are two samples within the upper 900 m of similar properties. However, higher insoluble particle (dust) content mainly correlates with the dominance of gypsum in comparison to other sulphates (see answer to Reviewer 1) and thus supports our interpretation regarding the deeper samples.

2) Change in mineralogy around 900 m

In the abstract we say "A variety of sulphates dominate the upper 900 m while gypsum is the only sulphate in deeper samples, which however contain more mineral dust, nitrates and dolomite." We thus not only refer to (the debatable distribution of) sulphates, but also to other minerals (e.g., nitrates and dolomite). These minerals only occur below 900 m while other minerals, such as hematite and titanite, only occur above 900 m.

All these results support our interpretation of a change at 900 m.
We weaken our statement throughout the text and e.g., delete "considerable change" in the abstract:
"Inclusions of the same composition tend to cluster, but clustering frequency and mineralogy changes with depth."
We also mention that the deepest two samples are difficult to compare with the samples above **(l. 498)**:
"However, it is difficult to compare the Holocene samples to the two samples from the Younger Dryas and Bølling Allerød. A systematic follow-up study on EGRIP Glacial ice is needed to investigate if the observed trends, e.g., of mineral diversity, continue with depth."

**Ref2:** l. 37: 'CFA' is unexplained at this point.
**Reply:** We added the explanation for CFA **(l. 42)**.

**Ref2:** l. 111: 'thin sections' is unexplained at this point.
**Reply:** We changed it to **(l. 132)**: "Depth co-registration to the samples analysed with Raman spectroscopy is limited by…".

**Ref2:** l. 290-294: This section appears to belong in the conclusions?
**Reply:** We partly agree with this statement. Referring to Eichler et al. (2019) and to the mentioned figures helps to convey the concept. We add a summary of this paragraph to the conclusions **(l. 512)**: "Combining these methods, and thus covering different scales, provides a good basis for a systematic analysis of different depth regimes while ensuring a sufficient number of micro-inclusions."

**Ref2:** l. 344: Does 'the stadial' refer to the Younger Dryas interval in this case?
**Reply:** This refers to the Greenland Stadial 1, i.e. the Younger Dryas. To clarify this we changed the text to **(l. 347)**: "Even though we find this difference in mineralogy between the upper and lower part, this change does not correlate with a major change of the climate period, i.e. from Glacial to Holocene ice."

**Ref2:** l. 445: What is Dome Fuji Interstadial ice?
**Reply:** This refers to the Dome Fuji ice core (introduced in **l. 351**) and the discussed differences between Holocene and interstadial ice by Ohno et al. (2005) on p. 176:"There are also small differences between Holocene and interstadial ice (1351 m)."

**Ref2:** l. 458-465: It seems unnecessary to repeat part of the introduction here.
**Reply:** We deleted l. 457-463.

**Additional note:** We found a mistake in our diversity index calculation, the numerator and denominator were exchanged. After correcting this a value of 1 means every inclusion has a different mineralogy, while small values indicate a low diversity. This led to changes in the figure (see figure below) and in the text:
"To compare our samples despite the varying amount of total identified Raman spectra per sample we calculated the ratio of the amount of different minerals per sample ($n_m$) to the total amount of identified micro-inclusions per sample ($n_i$) resulting in the diversity index Ivar with a maximum of 1. Ivar of 1 indicates that every inclusion is of different mineralogy while values close to 0 indicate a low diversity.
$$Ivar = \frac{n_m}{n_i}$$
Ivar varies between 0.099 and 0.308, the mean value is 0.158. Mineralogy diversity decreases slightly with depth (Fig. 6), the large diversity of sulphates is only found in the upper 900 m (Fig. 5B)."
"The lack of a variety of sulphate minerals below 900 m is shown in our diversity index, which decreases with depth even though other minerals occur at these depths, such as dolomite."

[Figure]

Figure 4 Mineral number and diversity with depth in EGRIP ice. A) Absolute numbers of different minerals per sample. The dotted blueline is the median value (10). B) Diversity index values calculated after Eq. (1). The light blue line is a linear regression, the dotted blue line is the mean value (0.158). Higher values indicate a larger mineral diversity in relationship to the amount of identified Raman spectra per sample.

---

## Referee Report (RR1)

This work investigates mineralogy of micro-inclusions in the EGRIP ice core with multiple methods, including Continuous Flow Analysis, optical microscopy and micro Raman spectroscopy.

As far as I know, this is one of the most detailed studies on the microstructural distribution of impurities in polar ice.

Because impurities are not only important climate proxies but also considered to be factors controlling various physical properties of polar ice such as ice deformation, results of the work will be interesting for many potential readers.

However, in the present state, it is not clear whether the main findings of this study (the diversity in mineralogy) is correct or not, due to the critical lack of information (results and discussion of Raman spectra).

Figure 3:

Raman shifts of the three spectra shown in the figure are considerably (about 5 cm$^{-1}$) larger than those reported in previous studies: for example, the position of the main peak for feldspar is 508 cm$^{-1}$ (e.g. Sakurai and others, 2011), that for gypsum is 1008 cm$^{-1}$ (e.g. Ohno and others, 2005; Sakurai and others, 2011), that for quartz is 465 cm$^{-1}$ (Sakurai and others, 2011). Although a small difference in Raman shift (a few cm$^{-1}$) between reports may be caused by differences in apparatus and/or calibration method, this discrepancy is too large. Information about how the apparatus was calibrated and the accuracy of Raman shift (spectral resolution) must be shown in the paper.

Figure 3:

My understanding is that one of the highlights of this work is to investigate the variety of minerals, including previously unreported species. However, Raman spectra being the key in analyzing the chemical forms of micro-inclusions are shown only for the three minerals in the paper. To verify the statements about the diversity in mineralogy, spectral information about the other minerals are needed.

Lines 205-206:"Sulphates can be difficult to distinguish due to tiny differences in Raman spectra, but most sulphates were identified (Fig. 4B)."

Indeed, it is difficult to distinguish some sulphates, especially between Mg-sulphate and Na-sulphates due to tiny (about 1cm-1) differences in the position of the main peaks (Ohno and others, 2005). Therefore, many earlier works (e.g. Ohno and others, 2006; Sakurai and others, 2011) hesitated to distinguish the two sulphates, and reported

peaks around 989-990 cm-1 as Mg-sulphates and /or Na-sulphates. I want to know how the authors told Mg-sulphates from Na-sulphates using the "compact" spectrometer (the spectra must be shown). As a rule, focal Length of a compact spectrometer is short, resulting in low spectral resolution.

Lines 207-208:" Spectra with a strong peak at 1050 cm−1 probably indicate K-nitrates while a strong peak around 1070 cm−1 might indicate Na-nitrates (Ohno et al., 2005)."
The direct comparison with the spectra by Ohno and others (2005) is not inappropriate. As mentioned above, for some reason, Raman spectra in the present work are shifted by approximately 5 cm-1 compared with those of the previous study.

Figure 4:
It is difficult to distinguish some minerals because of similar colors. In addition to different colors, the use of different patterns will be helpful to differentiate mineral species.

Line 233:"below a depth of"
Above?

Lines 304-305:" We identified most sulphates as gypsum or Na-sulphates, which agrees with e.g., Ohno et al. (2005); Eichler et al. (2019) (Fig. 4B)."
The statement of this sentence inconsistent with the Fig. 4B. The figure shows that Na-sulphates (yellow) is a minor species.

Line 479: "26 different spectra were identified"
All 26 spectra should be shown in the paper. Also, Raman peak assignment by comparing those with reference spectra must be done. First, the spectral information is essential to validate the identifications. Second, if the identifications are correct, the spectral information will be very valuable for potential readers.

---

## Author Response (AR2)

**Comments to the editor**:

Dear Dr. Sandells,

Thanks for noticing, there are indeed some mistakes regarding the mineral distribution. Due to the small number of changes, we do not provide an updated track-changes file right now, but could do so if needed. We address the changes as follows:

**Hematite (in Figure 5 S2 and S8-11) vs line 248 'Minerals such as hematite were only found in the samples below 900m' vs line 340 'Rare minerals, such as hematite, anatase and titanite were found independent of depths'**
We changed this to: (l. 248) „Minerals, such as epidote or jacobsite, were only found in the samples below 614 m.
(l. 340) 'Rare minerals, such as hematite, anatase and titanite were found independent of depths' is the correct statement in this case.

**Rutile (in Figure 5 S2 and S5) vs line 341 'Other minerals e.g., rutile and epidote, were only found in shallow samples' vs line 377 'while rutile, titanite, epidote, and jacobsite have only been found in one sample'**

The first statement is correct, Rutile was found in S2, S4, and S5; epidote in S4. The second sentence was thus changed to "Minerals, such as hematite and anastase, were only found in a few samples while epidote, prehnite, pyromorphite, and dolomite were only found in one sample."

**Please also correct typo on line 254 'one krohnkite micro-inclusion in S10': should be S9, and typo on line 321: 'gypusm' -> 'gypsum'**
Corrected to „gypsum".

**Line 251: 'The majority of sulphates was found in the seven shallowest samples below a depth of 900 m': Should this be above a depth of 900m?**
Indeed, we changed it to "above a depth of 900 m."

---

## Author Response (AR3)

**Ref:** I have read the revised version of "Microstructure, Micro-inclusions and Mineralogy along the EGRIP ice core - Part 2: Implications for paleo-mineralogy" and find it to be improved. Particularly, the more complete analysis of the mineralogy that is now found in the discussion is welcome. There are still a few points where I think further improvement can be made, which I detail below. I use the line numbers that appear in the track-changes version of the manuscript.

**Ref:** Line 74: I would strike "taken at arbitrary depths", sampling design is usually carefully thought out and very seldom "arbitrary".
**Answer:** "Arbitrary" was deleted.

**Ref:** Lines 78-91: This is a great amount of detail on a topic that is not directly relevant to this manuscript. It is good to give broader context, but the link between mineralogy and deformation structure is not subsequently addressed in this manuscript.
**Answer:** We keep this short part due to its importance for the general motivation and the linkage to part 1.

**Ref:** Line 101: "a profound understanding on…": reword
**Answer:** Changed to: However, to develop a deep understanding of the mineralogy and localisation of impurities in polar ice, a combination of both approaches is required."

**Ref:** Lines 108-111: In my previous review, I said that the introduction needed clearer research questions. But that doesn't mean these need to be literally posed as questions. An introduction needs to identify a key gap in knowledge and outline how the current effort will address that gap. This can usually be accomplished without directly using a question mark.
**Answer:** Changed to "We further investigate whether the mineralogy of the deposited impurities was stable over the last 14 ka or whether there were changes as a result of the evolution of the Greenland Ice Sheet. Deriving a mineralogy record of inclusions might also enable a better understanding of the aerosol content of the past. Finally, we investigate whether the location of the inclusion in the ice microstructure is related to its mineralogy."

**Ref:** Line 122: This is a weak ending to the introduction. Why put something that is not the main focus as your final introductory remark? End with the main thing you hope to accomplish. If you don't think mineral formation is highly relevant, bury it mid paragraph and say something like: "these analyses also offer limited insights into mineral formation pathways". But tell us what you are actually trying to discover.
**Answer:** Good point, we deleted the last sentence and edited a sentence in the paragraph:
"In this study we investigate the mineralogy of these visible micro-inclusions, which also offer a limited insight into mineral formation pathways."

**Ref:** Line 127: Replacing unique with exclusive doesn't really solve the problem. In the rebuttal, you said these were the first measurements in a deforming ice stream. Say that. Let us know that these are new in part because they have never before been done in ice that was actively deforming to this degree.
**Answer:** We changed it to: At the drill site the ice flows with a velocity of 55 m a$^{-1}$
(Hvidberg et al., 2020) offering a novel possibility to study ice rheology and physical parameters contributing to deformation, such as crystal preferred orientation and impurity content, in an active deformation regime, i.e. an ice stream.

**Ref:** Line 148: I still don't understand why the data from the soluble section are not included in this paper. I know the authors are planning another manuscript. But several outstanding questions about mineral formation pathways could be efficiently resolved by appeal to the to composition of the dissolved fraction. I.e. did the sulphates gain their cations from sea water or from dust, etc.
**Answer:** CFA - chemistry data are not available yet, as the processing, synchronization and dating is still in progress. Further, sulfate was not measured online, but is currently measured on high-resolution discrete samples - also work in progress.
The analysis and interpretation of CFA chemistry and ion chemistry is beyond the scope of this paper.
To clarify this we changed the wording throughout the manuscript and now emphasise the use of micro-particle data instead of CFA data.

**Ref:** Table 2: Make it clear that "sulphates" and "mica" are the sum of all sulphate and mica species, respectively. At first, I thought they could be unidentified sulphates.

**Answer:** We changed it to "all sulphates" and "all mica".

**Ref:** Line 342: Maybe: "Because the EGRIP is uniquely located on an ice stream, and …"
**Answer:** Changed to: Because the EGRIP drilling site is uniquely located on an ice stream, and the assumed high impact of high-impurity layers on the deformation of the ice…

Line 344: I am still not too keen on these vague commands about what sort of future research is needed.
**Answer:** The addressed paragraph does not mention future research (in the new and track changes document), but summarises the advantages of our presented method.
This might refer to section 4.2.4 where we deleted the last sentence, which is indeed rather vague.

**Ref:** Line 363: The authors don't seem to appreciate just how rare. Kröhnkite is only known in 31 locations globally; most of these are copper ore bodies. A brief search reveals that it has been attested in one instance as a fumarolic mineral in Icelandic volcanism, but it is described as very rare in that setting too. 27 out of 386 sulphate minerals is 7%. The global abundance of kröhnkite as a fraction of sulphate minerals is many orders of magnitude below 7%. My first inclination is to assume that the authors have misidentified a more common mineral, but I am not enough of a Ramen expert to know how likely this is. If the authors are confident that their spectroscopy is correct, this is in fact a major mineralogical discovery. Given the mineral's rareness, it is very unlikely to be detrital and almost certainly formed in the snow or ice. But the authors need to think through how that could be possible. What is the total mass of copper implied? Is that even plausible? Etc.
**Answer:** The structure and general properties of Kröhnkite are very similar to bloedite, their main peak is thus also very similar ($\sim$1cm$^{-1}$ difference). The valid concern about the scarcity of Kröhnkite in general and the fact that is has not been observed in ice cores, make it very likely that these inclusions are also bloedite. As explained in more details in the answer to referee 2 we thus choose a conservative approach and classify the 27 kröhnkite inclusions as bloedite. Bloedite is much more common and has been observed before with Raman spectroscopy (Eichler et al., 2019). We appreciate the consist input of the referee regarding this matter.

**Ref:** Line 378: "As displayed in…", just write "(Fig. 6.)".
**Answer:** Adapted.

**Ref:** Line 382: What does "partially high" mean?
**Answer:** This indicated that gypsum, at some depths, is one of the most abundant minerals. We changed the text to "We have observed a pronounced change in mineralogy from a sulphate-diverse, while terrestrial dust rich, regime in the upper 900 m to a terrestrial dust-dominated regime with comparably large numbers of gypsum at some depths (e.g., S8 and S11)".

**Ref:** Line 384: "other minerals", you mean new minerals or minerals not found higher in the core, right?
**Answer:** Yes correct, we changed this to: The lack of a variety of sulphate minerals below 900 m is shown in our diversity index, which decreases with depth even though some minerals, such as dolomite, only occur below 900 m.

**Ref:** Line 385: "Varying numbers" is an incredibly vague way of putting it.
**Answer:** Defining the numbers more precisely is almost impossible here since we refer to three studies (with different absolute amounts of identified inclusions) and the two main mineral groups, i.e. sulphates and mineral dust, consisting of several minerals. Describing the exact amounts of these minerals results in a lengthy and confusing description of the work of others, which is not needed here.

**Ref:** Line 388: By rare, you mean uncommon in your ice core, not globally rare. I wouldn't even call hematite rare at n=33.
**Answer:** True, we changed it to "comparably rare minerals, such as…"

**Ref:** Line 389: maybe, " Hematite … abundance does not detectably vary with depth"
**Answer:** Changed to: The abundance of comparably rare minerals, such as hematite, anatase and titanite, does not detectably vary with depth.

**Ref:** Line 395: Maybe, "Though previous data is scarce, Sakurai…"
**Answer:** Adapted.

**Ref:** Line 397: The topic sentence doesn't match well with the rest of the paragraph. If your differences don't correspond to glacial-interglacial transitions, why go on about other workers who do? You can discuss both, but it needs to be more clearly framed. Something like, "the changes we see at 900 m are similar to that observed in other settings at a glacial-interglacial transition".

**Answer:** Changed to: „The changes in mineralogy that we observe at 900 m depth are similar to observations made in other settings at a transition

between Glacial and Interglacial. Eichler et al. (2019) found a strong difference in mineralogy..."

**Ref:** Line 452: Carbonate can also form in situ, as a chemical weathering product of carbonic acid interactions with mineral dust.

**Answer:** Thanks for noticing, we deleted carbonates here.

**Ref:** Line 461: Delete "it is suggested that"

**Answer:** Deleted.

**Ref:** Line 465: This is where the rest of the CFA data would be extremely helpful. And you apparently do have CFA data for H+ (figure 1), suggesting quite a bit what is very likely HCl (though probably other anions too) in most of the ice core. And indeed the portion with the dolomite (S10) is the one the part that is not acidic. This suggests either that carbonate is dissolving in the other portions of the core or that it precipitated in the part where acidity was low. In other instances, do the changes in sulphate abundance and mineralogy correlate with these changes in acidity? It is well worth doing some more analysis here.

**Answer:** The presented H+ acidity data is from published ECM measurements (Mojtabavi et al., 2020). The major feature in the acidity record is the drop in the Younger Dryas, variations in acidity are much smaller during the Holocene and Bolling Allerod. Acidity is slightly lower between 350 and 925 m, but a significant change in mineralogy in this depth regime was not observed (compared to above 350 m and below 925 m). We provided a high-resolution analysis of the acidity at the exact sample depths in the first review round and did not find any correlations regarding the mineralogy content.

We extend section 4.3.4 rare minerals with a brief summary on dolomite:

"Dolomite was only observed in S10 originating from the Younger Dyas, which is the only section of the ice core that is not acidic (Fig. 1). This indicates either that the carbonate is dissolving in the other parts of the core or that it has precipitated in the part where the acidity was low."

**Ref:** Line 480: Do your results really show this? There is a major outlier with the Younger Dryas. Though I suppose the word "often" accounts for this. I think the diversity in sulphate mineralogy requires both acidity and dust. Perhaps a multiple regression (or another similar statistical test) is in order here.

**Answer:** Our results show, that the only sulphate in the mentioned sample is gypsum (Fig. 5b) while dust values are very high (Fig. 1). This supports our statement that gypsum dominates over other sulphate minerals when dust values are high. We agree that more parameters are needed for a diverse mineralogy than dust alone, but we cannot discuss the impact of acidity or conductivity in detail here. A multiple regression, or a similar test, is indeed an interesting approach and will be especially useful when it is possible to implement data from the Glacial. Analysing the relationship between different environmental parameters is beyond the scope of this manuscript.

**Ref:** Line 483: Nitrate salts are a not common terrestrial mineral, being found mostly in extremely arid environments. I think it is a mistake to presume detrital origin here. Nitric acid (as N2O5 or HNO3) is known the be a component of atmospheric fallout and therefore of ice cores. It is far more likely that the nitrate salts formed in some sort of competition with the sulphosalts, particularly since they are found where sulphate abundance is low.

**Answer:** We discuss this process a few lines further down, but adapted the paragraph in the following way:

"Nitrates are well soluble in water and a major impurity component in polar ice as obtained from CFA (Röthlisberger et al., 2000b) and IC (Eichler et al., 2019) analyses, but there is a lack of understanding in which form they are present in ice. Nitric acid (N2O5 or HNO3) is a component of atmospheric fallout, and thus of ice cores, and could compete with other acids, such as sulphuric acids. Sulphuric acid competes with other acids to react with the relative rare cations, replaces other acids in their salts and thus forms a variety of sulphate salts (Iizuka et al., 2008). The relative abundance of nitrates at a certain depth (Fig. 5A) indicates that similar processes occur with nitrates. However, nitrate ions seem to be more likely to exist in dissolved forms than in particle forms due to their good solubility in water."

**Ref:** Line 490: Epidote and prehnite are both fairly common metamorphic minerals, so I don't think they are overrepresented in your core. Pyromorphite is genuinely rare, but as you observe, only shows up once.
**Answer:** No changes.

**Ref:** Line 501: Jacobsite is the tricky one as there is now a "body of literature" suggesting that could actually form in ice cores, but I think you handle this fine.
**Answer:** No changes

**Ref:** Line 506: I would delete this final sentence. I think you've said enough given the data you have.
**Answer:** Adapted, we deleted the sentence.

**Ref:** Line 509: I think now that you've added all this additional analysis of the mineral formation pathways above, this no longer needs to be framed as "another possibility".
**Answer:** Changed to: "Chemical reactions occurring inside the ice could explain the diversity in chemical compounds as summarised for snow by Bartels-Rausch et al. (2014), and further discussed for ice by Steffensen (1997); de Angelis et al. (2013); Baccolo et al. (2018) and Eichler et al. (2019)."

**Ref:** Line 529: This is very intriguing. The notion that the formation of clathrates juggles around the salt chemistry is, as far as I know, novel and certainly a possible explanation of why you loose sulphates and gain nitrates.
**Answer:** No changes.

**Ref:** Line 530: What is a "prominent diversity"?
**Answer:** We deleted "prominent", this is indeed confusing wording: "We observed a diverse mineralogy of micro-inclusions in EGRIP ice across all scales."

**Ref:** Line 547: This should use the standard terminology of non-seasalt-sulphate that is used elsewhere in the ice core literature. It should also be clearly explained what climatic factors would drive changes in the ratio of sea salt to non-seasalt-sulphate. Again, if they included the CFA data, the authors would be able to calculate non-seasalt-sulphate abundance exactly and see if this mineralogical shift really is driven by changes in sulphate source.
**Answer:** Changed to "…impacting the input of sea salt (ss), a major source of sulphates (ss–$SO_4^{-2}$). The variability in sulphate diversity and number could display the variety in blown-in sea salt aerosols (ssa) carrying sulphate compounds, such as gypsum."
A more detailed description of the processes is not possible yet, because the data is still being processed while samples are still being measured (see answer above).

**Ref:** Line 566: Aerosols are not the same thing as dust. The dust is mostly from Asia, but the aerosols have far more diverse and complex origins.
**Answer:** True, we misused aerosols here and changed it to "Dust particles deposited in the Holocene are…".

**Ref:** Line 618: You might as well add the bubble-clathrate transition, since you are listing a number of other speculative possibilities.
**Answer:** Changed to: "the varying sea-ice cover impacting the availability of sulphates in the air during deposition or processes taking place at the bubble-clathrate transition."

This work investigates mineralogy of micro-inclusions in the EGRIP ice core with multiple methods, including Continuous Flow Analysis, optical microscopy and micro Raman spectroscopy.
As far as I know, this is one of the most detailed studies on the microstructural distribution of impurities in polar ice.

Because impurities are not only important climate proxies but also considered to be factors controlling various physical properties of polar ice such as ice deformation, results of the work will be interesting for many potential readers.
However, in the present state, it is not clear whether the main findings of this study (the diversity in mineralogy) is correct or not, due to the critical lack of information (results and discussion of Raman spectra).

**Ref:** Figure 3:

Raman shifts of the three spectra shown in the figure are considerably (about 5 cm$^{-1}$) larger than those reported in previous studies: for example, the position of the main peak for feldspar is 508 cm$^{-1}$ (e.g. Sakurai and others, 2011), that for gypsum is 1008 cm$^{-1}$ (e.g. Ohno and others, 2005; Sakurai and others, 2011), that for quartz is 465 cm$^{-1}$ (Sakurai and others, 2011). Although a small difference in Raman shift (a few cm$^{-1}$) between reports may be caused by differences in apparatus and/or calibration method, this discrepancy is too large. Information about how the apparatus was calibrated and the accuracy of Raman shift (spectral resolution) must be shown in the paper.

**Answer:** We used the same system as Weikusat et al. (2012, 2015) and Eichler et al. (2019) and calibrated it using the standard procedure deployed by the manufacturer WITec. Thus, a Hg/Ar spectral calibration lamp was connected to the spectrograph with a 25 µm fiber cable enabling the automatic detection of known sharp peaks followed by an automated calibration. The applied grating of 600 enables a spectral range of >3700 cm$^{-1}$ with a pixel resolution of <3 cm$^{-1}$ / <0.025 nm. We implement this information in the paper now:
"Spectroscopy analysis was performed at the Alfred Wegener Institute Helmholtz Centre for Polar- and Marine Research, Bremerhaven with a WITec alpha 300 M+ combined with a NdYAG laser (λ = 532nm) and a UHts 300 spectrometer with a 600 grooves mm-1 grating resulting in a spectral range of >3700 cm$^{-1}$ and a pixel resolution of <3 cm$^{-1}$."

The identification of the minerals was done by comparing our spectra with data from the largest open-access data base for Raman spectroscopy: the RRUFF data base. Depending on the exact chemical composition of the mineral spectra can differ slightly even though the mineral is the same. Our Raman shifts are indeed slightly larger than the ones shown by Ohno et al. (2005) and Sakurai et a., (2011), both used the same Raman system. Our spectra are similar to the ones measured by Eichler et al. (2019), which used the same system as we did. However, since several peaks are usually used for the identification of a spectrum, small differences in Raman shift are tolerable (and normal) between different systems and/or mineral composition.
Unfortunately, the spectra in Fig. 3 were chosen too hastily, the observed Raman shifts are usually lower than the values in Fig. 3 (more accurate are e.g., quartz = 466 cm$^{-1}$, gypsum = 1012 cm$^{-1}$ and 1140 cm$^{-1}$). The spectra are thus within the variety range between different instruments and boundary conditions. We chose more accurate spectra for Fig. 3 and implement a figure showing an overview of all spectra in the appendix.

**Ref:** Figure 3:
My understanding is that one of the highlights of this work is to investigate the variety of minerals, including previously unreported species. However, Raman spectra being the key in analyzing the chemical forms of micro-inclusions are shown only for the three minerals in the paper. To verify the statements about the diversity in mineralogy, spectral information about the other minerals are needed.
**Answer** : This figure is provided in the appendix now.

**Ref:** Lines 205-206:"Sulphates can be difficult to distinguish due to tiny differences in Raman spectra, but most sulphates were identified (Fig. 4B)."
Indeed, it is difficult to distinguish some sulphates, especially between Mg-sulphate and Na-sulphates due to tiny (about 1cm-1) differences in the position of the main peaks (Ohno and others, 2005). Therefore, many earlier works (e.g. Ohno and others, 2006; Sakurai and others, 2011) hesitated to distinguish the two sulphates, and reported peaks around 989-990 cm-1 as Mg-sulphates and /or Na-sulphates. I want to know

how the authors told Mg-sulphates from Na-sulphates using the "compact" spectrometer (the spectra must be shown). As a rule, focal Length of a compact spectrometer is short, resulting in low spectral resolution.

**Answer:**
This was indeed a point of major discussion throughout the entire process. We now decided to follow the more conservative approach and refrain from differentiating between these two sulphates in such detail. In the results we now classify them (following Ohno et al. (2005), Sakurai et al. (2011)) as Mg- and/or Na-sulphates. We now also apply the same approach to kröhnkite and bloedite (see also answer to referee 1), because these two minerals have similar lattices and thus Raman spectra peaks. Especially when measuring inclusions in ice, i.e. having an overlaying ice spectra, this is the safer approach.
This lowers the absolute number of identified minerals to 24 and has a slight influence on the diversity index (new Fig. 6 below).

[Figure]

**Ref:** Lines 207-208:" Spectra with a strong peak at 1050 cm−1 probably indicate K-nitrates while a strong peak around 1070 cm−1 might indicate Na-nitrates (Ohno et al., 2005)." The direct comparison with the spectra by Ohno and others (2005) is not inappropriate. As mentioned above, for some reason, Raman spectra in the present work are shifted by approximately 5 cm-1 compared with those of the previous study.
**Answer:** Our wording here is unprecise und thus confusing. We observed pronounced peaks at 1051-1052 cm$^{-1}$ and 1070 cm$^{-1}$. Considering our slight shift in wavenumber compared to Ohno et al. (2005), their values of 1050 cm$^{-1}$ and 1068 cm$^{-1}$ for Na-nitrates and K-nitrates, respectively, are thus appropriate to use.
We changed the text to:
Spectra with a peak at 1051-1052 cm$^{-1}$ indicate K-nitrates while a peak at 1070 cm$^{-1}$ indicates Na-nitrates (slightly shifted compared to Ohno et al., 2005).

**Ref:** Figure 4:
It is difficult to distinguish some minerals because of similar colors. In addition to different colors, the use of different patterns will be helpful to differentiate mineral species.
**Answer:** We tried different approaches and several patterns turned out to be more confusing, especially when inclusions are very close to each other. We thus decided to keep the (colour-blind friendly) colours. Since we now differentiate between two less sulphates it should be easier to distinguish them (see Fig. 5, 7, 8 , 9).

**Ref:** Line 233:"below a depth of" Above?
**Answer:** In the latest version this was already changed to "above".

**Ref:** Lines 304-305:" We identified most sulphates as gypsum or Na-sulphates, which agrees with e.g., Ohno et al. (2005); Eichler et al. (2019) (Fig. 4B)."
The statement of this sentence inconsistent with the Fig. 4B. The figure shows that Na-sulphates (yellow) is a minor species.
**Answer:** Thanks for noticing, Mg-sulphates were meant and the text was changed according to the reply above to "Na- and Mg-sulphates".

**Ref:** Line 479: "26 different spectra were identified"

All 26 spectra should be shown in the paper. Also, Raman peak assignment by comparing those with reference spectra must be done. First, the spectral information is essential to validate the identifications. Second, if the identifications are correct, the spectral information will be very valuable for potential readers.

**Answer:** A figure in the appendix provides the Raman spectra. Reference spectra are openly accessible in e.g., the RRUFF database or commercial databases.